# Infection- or AZD1222 vaccine-mediated immunity reduces SARS-CoV-2 transmission but increases Omicron competitiveness in hamsters

Julia R. Port [1,7], Claude Kwe Yinda[1,7], Jade C. Riopelle[1], Zachary A. Weishampel[1], Taylor A. Saturday[1], Victoria A. Avanzato[1], Jonathan E. Schulz[1], Myndi G. Holbrook [1], Kent Barbian[2], Rose Perry-Gottschalk [3], Elaine Haddock [1], Craig Martens[2], Carl. I. Shaia [4], Teresa Lambe [5,6], Sarah C. Gilbert [5], Neeltje van Doremalen [1,8] & Vincent J. Munster [1,8] ✉

Limited data is available on the effect of vaccination and previous virus exposure on the nature of SARS-CoV-2 transmission and immune-pressure on variants. To understand the impact of pre-existing immunity on SARS-CoV-2 airborne transmission efficiency, we perform a transmission chain experiment using naïve, intranasally or intramuscularly AZD1222 vaccinated, and previously infected hamsters. A clear gradient in transmission efficacy is observed: Transmission in hamsters vaccinated via the intramuscular route was reduced over three airborne chains (approx. 60%) compared to naïve animals, whereas transmission in previously infected hamsters and those vaccinated via the intranasal route was reduced by 80%. We also find that the Delta B.1.617.2 variant outcompeted Omicron B.1.1.529 after dual infection within and between hosts in naïve, vaccinated, and previously infected transmission chains, yet an increase in Omicron B.1.1.529 competitiveness is observed in groups with pre-existing immunity against Delta B.1.617.2. This correlates with an increase in the strength of the humoral response against Delta B.1.617.2, with the strongest response seen in previously infected animals. These data highlight the continuous need to improve vaccination strategies and address the additional evolutionary pressure pre-existing immunity may exert on SARS-CoV-2.

There is limited evidence on the effects of vaccination on SARS-CoV-2 transmission. Reduction in transmission in households has been documented[1,2] and reduced viral load in the upper respiratory tract of infected individuals has been demonstrated[3,4].

Ongoing evolution in the human population resulted in the emergence of variants of concern (VOCs). Phenotypic changes that characterize VOCs are increase in transmissibility, increase in virulence, change in clinical disease presentation, and/or decrease in effectiveness of public health and social measures or available diagnostics, vaccines, and therapeutics[5,6]. Changes in the transmission phenotype can occur by a variety of adaptions including virus shedding dynamics, human behavior, host cell tropism, and entry. Furthermore, a large portion of the human population is no longer naïve to SARS-CoV-2[7–9]. Immunity induced by previous exposure or

vaccination have changed the susceptibility to infection and thus the evolutionary pressures on SARS-CoV-2.

The emergence of VOCs is following almost a classic pattern in which the new VOC replaces the old VOC: this was observed for Alpha, Delta, and now Omicron. Whereas the initial replacements of previous VOCs by a new variant were due largely to an increase in the transmission potential of the virus[10–14], the transmission advantage of Omicron over Delta in humans is not fully understood[15]. Due to antigenic differences, the anti-Omicron cross-reactivity of neutralizing antibodies mounted against other variants is low[16–21].

The absence of infectious virus shedding kinetics and epidemiologic data about transmission between previously infected, vaccinated, and unvaccinated individuals with SARS-CoV-2 make it difficult to determine whether breakthrough infections have the potential to contribute to the spread of SARS-CoV-2[22], especially in the context emergence of the Omicron variants. Compared to Delta, Omicron is more likely to cause infections in a vaccinated population[23]. To better understand SARS-CoV-2 evolution, it will be crucial to differentiate between two separate evolutionary pressures: increasing transmissibility and antigenic escape.

Previously, we have experimentally shown an increased aerosol transmission phenotype of SARS-CoV-2 Alpha over Lineage A[24,25]. In this work, we are using infection- or vaccine-mediated immunity to model the impact of this evolutionary pressure on the transmission of Delta B.1.617.2 and Omicron B.1.1.529. We find a clear gradient in transmission blockage; intranasal vaccination and previous infection result in higher reduction in transmission compared to intramuscular vaccination. While the Delta B.1.617.2 variant outcompetes Omicron B.1.1.529 after dual infection regardless of pre-existing immunity status, Omicron B.1.1.529 competitiveness increases in correlation with an increase in the strength of the anti-Delta B.1.617.2 humoral response.

## Results

### Previous exposure or vaccine-induced pre-existing immunity reduces virus replication, shedding, and lung pathology after reinfection

We compared the impact of pre-existing immunity on attenuating disease after SARS-CoV-2 infection. Pre-existing immunity was achieved by intranasal (IN) or intramuscular (IM) vaccination with AZD1222 (against Lineage A), or previous infection (PI) with the antigenically close Delta B.1.617.2[26]. Six hamsters per group were immunized with AZD1222 (ChAdOx1 nCoV-19, $2.5 \times 10^8$ IU/animal) or exposed via direct contact to IN-inoculated animals one day after inoculation (5:1 sentinel: donor ratio). In all vaccinated and PI animals, seroconversion was confirmed after at least 21 days. At least 35 days after immunization via vaccination or infection, animals were challenged via the IN route using $10^4$ TCID$_{50}$ SARS-CoV-2. Naïve, age-matched hamsters served as a control group. We measured sgRNA, which is a surrogate for virus replication[27,28], in nasal turbinates and lungs at day 5 post challenge. In naïve animals, virus replication was observed in nasal turbinates (median = 6.873 sgRNA copies/gr (Log$_{10}$)) and lung tissue (median = 8.303 sgRNA copies/gr (Log$_{10}$)). In contrast, viral RNA load was significantly reduced or absent in IM, IN, and PI groups as compared to naïve donors (Kruskal–Wallis test, followed by Dunn's multiple comparison test, $N = 6$; nasal turbinates: $p = 0.1479$, 0.0081, 0.0117, respectively; lung: $p = 0.0010$, 0.0069, 0.0010, respectively). In lung tissue, sgRNA was only detected in 1 out of 6 animals in the IN group (4.93 sgRNA copies/gr (Log$_{10}$)), but not in the other groups. sgRNA was detected in 3 out of 6 nasal turbinate samples in the IM donors (median = 2.581 sgRNA copies/gr (Log$_{10}$)), 1 out of 6 in the IN group (4.423 sgRNA copies/gr (Log$_{10}$)), and 2 out of 6 in the PI group (median = 1.173 sgRNA copies/gr (Log$_{10}$)), Fig. 1a, b).

Vaccination and previous infection reduced overall respiratory shedding. We measured sgRNA on 2, 3, and 5 days post inoculation (DPI) in oral swabs. Cumulative virus burden (area under the curve

(AUC)) in oral swabs was marginally reduced after IM vaccination (median AUC (Log$_{10}$) = 19,489, $p = 0.999$, $N = 6$, Kruskal–Wallis test, followed by Dunn's multiple comparison test), moderately reduced after IN vaccination (median AUC (Log$_{10}$) = 13,470, $p = 0.4347$), and significantly reduced in the PI group (median AUC (Log$_{10}$) = 454.4, $p = 0.0197$), compared to naïve animals (median AUC (Log$_{10}$) = 43,618) (Fig. 1c).

We compared the severity of lung disease as measured by the lung:body weight ratio (Fig. 1d). In the donor hamsters, previously established immunity reduced the lung:body weight ratio significantly after challenge (naïve = 1.296, IM = 0.7343, IN = 0.8030, PI = 0.8077, $N = 6$, Kruskal–Wallis test, followed by Dunn's multiple comparison test, run against the naïve group, $p = 0.021$, $p = 0.0165$, and $p = 0.0383$, respectively). Hamsters from the naïve group developed lesions typical of SARS-CoV-2 in this model[26] (Fig. 1e, Table S1). SARS-CoV-2 nucleoprotein immunoreactivity ranged from moderate to numerous in both bronchi and alveoli and was especially apparent at the periphery of foci of pneumonia (Fig. 1e, g). CD3 immunoreactivity, a measurement of T-cell infiltration, was greatly increased in foci of inflammation and pneumonia in the lung (Fig. 1f). IM vaccination decreased the disease severity, as previously described[29,30], which was accompanied by decreased antigen presence and T-cell infiltration compared to naïve animals. The majority of CD3 immunoreactive T-cells were located adjacent to bronchioles and blood vessels. In contrast, pathology in the IN vaccinated and PI hamsters was negligible and limited to scant inflammation and terminal airway reactivity, with no detectable virus presence and consistently lower T-cell numbers than the naïve animals. No difference in B-cell infiltration was observed, as measured by PAX5 staining between the groups (Table S1). This was accompanied by decreased gene expression levels for interferon (IFN)γ and interleukin (IL)-10 in both the upper and lower respiratory tract, and IL-6 in the lower but not the upper respiratory tract, as compared to naïve animals. Expression levels of tumor necrosis factor (TNF)α remained unchanged. A trend towards increased IL-4 expression was observed in the upper respiratory tract, especially in PI animals, as compared to naïve controls (Fig. S2).

### Contact and airborne transmission in naïve Syrian hamsters

To establish the ability and limitations of transmission over multiple successful rounds through the air and through contact in the Syrian hamster model, we performed contact and airborne transmission chain experiments over two or three generations (1:1 ratio between donors and sentinels) and repeated these chains three times (Fig. 2a). Donors were intranasally inoculated with SARS-CoV-2 (1:1 mixture of Delta B.1.617.2 and Omicron B.1.1.529). One day later, generation 1 sentinels (sentinels 1) were exposed to the donors for 48 h, followed by exposure of sentinels 2 to sentinels 1 for 48 h, and finally exposure of sentinels 3 to sentinels 2 for 72 h. Each exposure was started on 2 DPI/DPE relative to the previous chain. Oropharyngeal swabs were collected from all animals at 2, 3, and 5 DPI/DPE, and lung and nasal turbinate samples were harvested at 5 DPI/DPE. To be certain animals sustained infection, we considered animals only infected when at least 2 out of 5 samples collected had detectable sgRNA (>10 copies/reaction (rxn)). In the direct contact chains, all animals became infected (Fig. 2a). In contrast, 2 out of 3 of the sentinels 1 and sentinels 2 hamsters, and 1 out of 2 sentinels 3 hamsters became infected in the airborne chains.

### Pre-existing immunity protects against contact and airborne transmission

To test whether pre-existing immunity would reduce the transmissibility of SARS-CoV-2, groups of animals with vaccine- or infection-induced pre-existing immunity were used in a contact and airborne transmission experiment (Fig. 2b). 16 hamsters per group were immunized with AZD1222 (ChAdOx1 nCoV-19, $2.5 \times 10^8$ IU/animal)

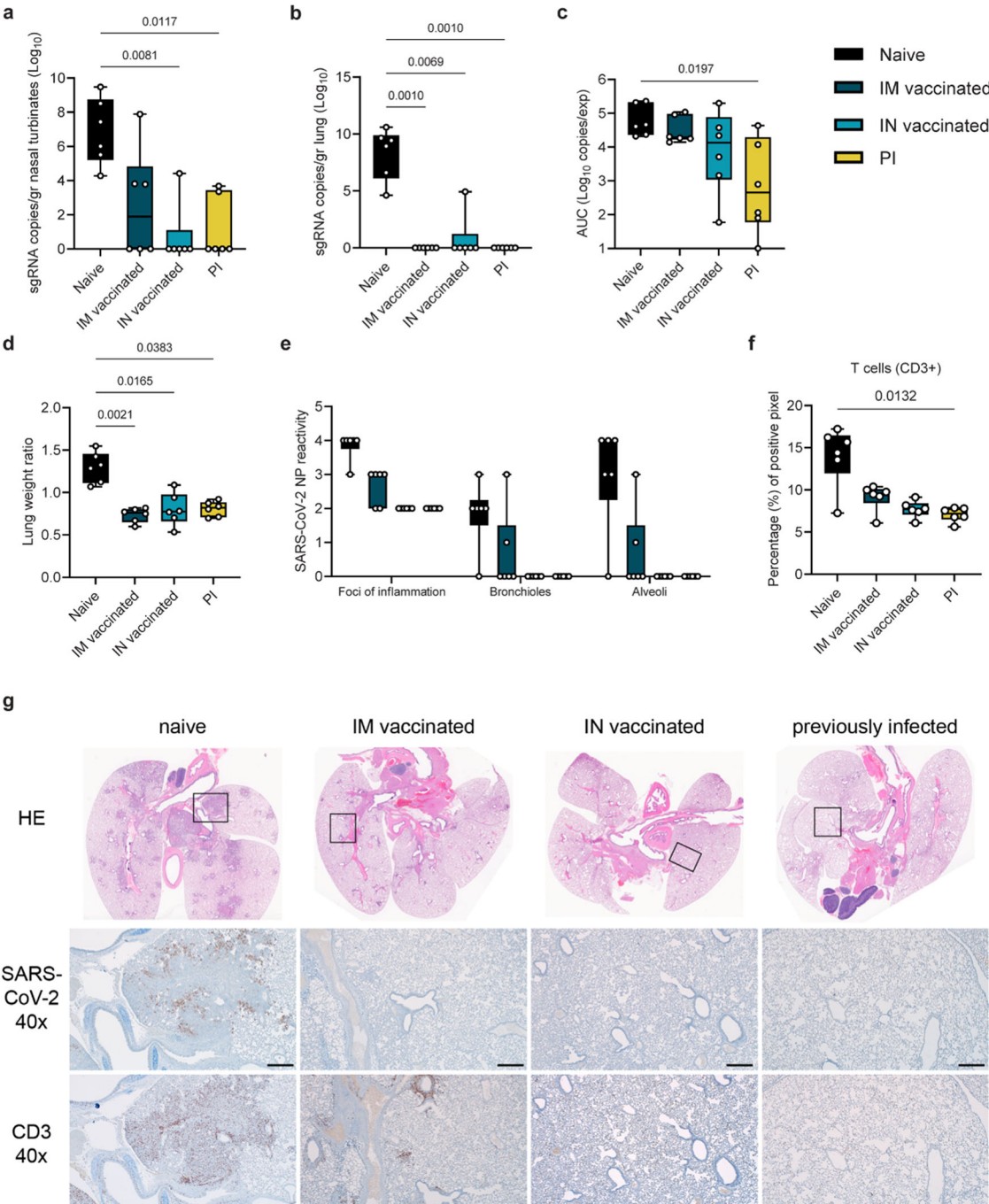

**Fig. 1 | Reduction of disease severity and shedding through pre-existing immunity.** Hamsters were either vaccinated intranasally (IN) or intramuscularly (IM) against Lineage A or experienced a previous infection (PI) with Delta B.1.617.2. Animals were inoculated with SARS-CoV-2 at least 28 days later through the intranasal route ($N = 6$). Tissue samples were collected at day 5. subgenomic (sg)RNA in nasal turbinates (**a**) and lungs (**b**). Whisker-plots depicting median, min and max values, and individual values, Kruskal–Wallis test, followed by Dunn's multiple comparison test. **c** Cumulative shedding. Area under the curve (AUC) of sgRNA measured in oral swabs taken on 2, 3, and 5 DPI (sgRNA copies/rxn/experiment). Whisker-plots depicting median, min and max values, and individual values, Kruskal–Wallis test, followed by Dunn's multiple comparisons test. **d** Lung weights (lung:body weight ratio). Whisker-plots depicting median, min and max values, and individual values, Kruskal–Wallis test, followed by Dunn's multiple comparison test.

**e** SARS-CoV-2 reactivity measured by immunohistochemistry (IHC) targeting SARS-CoV-2 nucleoprotein (NP) in upper and lower respiratory tract. Nucleoprotein reactivity score: 0 = none, 1 = rare/few, 2 = scattered, 3 = moderate, 4 = numerous, 5 = diffuse. Whisker-plots depicting median, min and max values, and individual values, Kruskal–Wallis test, followed by Dunn's multiple comparisons test. **f** T-cell infiltration into the lung, measured by CD3 antigen presence and positive pixel quantification. Whisker-plots depicting median, upper and lower quantile, min and max values, and individual values, Kruskal–Wallis test, followed by Dunn's. black = naïve, dark blue = IM vaccinated, light blue = IN vaccinated, yellow = PI. *P*-values stated were significant (<0.05). **g** Lung pathology. top = HE stains, middle = IHC for nucleoprotein, bottom = IHC for CD3. Squares indicate area of magnification. Scale bar: 500 μm at 40x magnification. Source data are provided as a Source Data file.

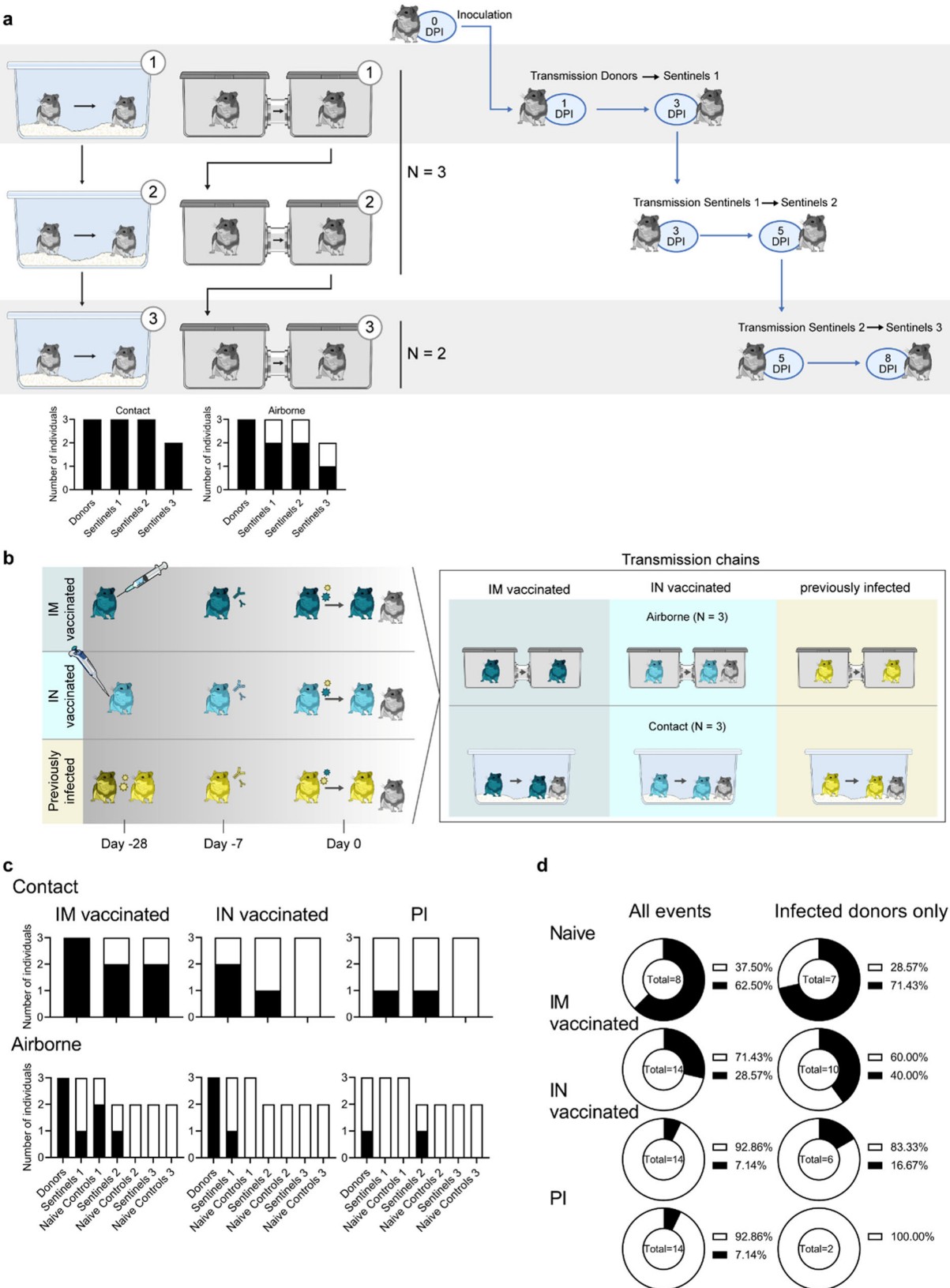

either IN or IM or were exposed via direct contact to IN-inoculated animals one day after inoculation (4:1 naive to infected ratio). Seroconversion was assessed at least 21 days after. Six animals (donors) were then challenged after at least 35 days (Delta B.1.617.2: Omicron B.1.1.529 mixture at a 1:1 ratio, total of $10^4$ $TCID_{50}$). Animals were considered infected if 2 out of 5 samples (either a swab,

nasal turbinates, or lung tissue sample) had detectable sgRNA (>10 copies/rxn). All IM vaccinated donors became infected. In contrast, 5 out of 6 donors in the IN vaccinated group, and 2 out of 6 donors in the PI group became infected (Fig. 2c). Donors were randomly assigned to a contact or transmission chain ($N = 3$ for each transmission route).

**Fig. 2 | Transmission competitiveness of Delta B.1.617.2 and Omicron B.1.1.529 in a naïve, vaccinated, or previously infected hamster population.**
**a** Transmission efficiency in naïve hamsters; schematic. Summary of infection status for the donors and sentinels. Oropharyngeal swabs were taken on 2, 3, and 5 days post infection/exposure (DPI/DPE), and lungs and nasal turbines were collected at day 5 DPI/DPE. Bar charts depict summary of individuals, divided into contact and airborne chains. **b** Transmission efficiency in hamsters with pre-exiting immunity, schematic. Hamsters were either vaccinated intramuscularly (IM, dark blue) or intranasally (IN, light blue) against Lineage A or experienced a previous infection with Delta B.1.617.2 through contact exposure to IN inoculated hamsters (PI, yellow). Donor animals ($N = 6$ for each group) were inoculated with a total of $10^4$

$TCID_{50}$ of Delta B.1.617.2 and Omicron B.1.1.529 via the IN route (1:1 ratio), and sentinels 1 ($N = 6$) were exposed 24 h later. **c** Summary of infection status for donors and sentinels. Oropharyngeal swabs were taken on 2, 3, and 5 DPI/DPE, and lungs and nasal turbinates collected at 5 DPI/DPE. Bar charts depict summary of individuals, divided by contact and airborne chains. **d** Pie charts summarizing transmission efficiency between naïve, IM vaccinated, IN vaccinated, and PI hamsters across all possible airborne transmission events (left) and events for which the donor animal was confirmed infected (2 out of 5 samples positive by sgRNA qRT PCR (>10 copies/rxn)) (right). Number of events is indicated within each pie chart. Pie chart colors: Black = transmission, white = no transmission. Source data are provided as a Source Data file.

For contact transmission, 24 h after SARS-CoV-2 challenge, donors were co-housed with one naïve sentinel and one immunized sentinel (sentinels 1, 1:1:1 ratio) for 48 h (Fig. S1, Table S2). For IM vaccination, 2 out of 3 naïve sentinels and 2 out of 3 immunized sentinel 1 hamsters became infected (Fig. 2c). Contact transmission was further reduced in the IN vaccinated and PI groups; only 1 out of 3 immunized sentinels 1 and no naïve sentinels 1 were infected (Fig. 2c).

Reduction in transmission was more prominent for the airborne route. Only 1 out of 3 immunized sentinels 1 was infected in the IM airborne chains, while 2 out of 3 naïve sentinel hamsters became infected. In the IN vaccinated airborne chains, 1 out of 3 immunized sentinels 1 and no naïve sentinel 1 hamsters were infected. In the PI chains, no immunized or naïve sentinel 1 became infected (Fig. 2c).

Due to the importance of airborne transmission, we decided to take two airborne transmission experiments per group out to sentinels 3 (as described above for the naïve hamsters: donors → sentinels 1 → sentinels 2 → sentinels 3). In the IM vaccinated group, 1 out of 2 immunized sentinel 2 animals, but no naïve sentinel 2, no immunized sentinel 3, and no naïve sentinel 3 were infected. In the IN vaccinated group, no sentinel 2 and no sentinel 3 became infected. In the PI group, 1 out of 2 immunized sentinel 2 animals, but no naïve sentinel 2, nor any sentinel 3 became infected.

We compared the airborne transmission efficiency between naïve, IM vaccinated, IN vaccinated, and PI hamsters for transmission events, where the donor animal was confirmed to be positive. We included immunized and naïve sentinels. For naïve hamsters ($N = 7$ events with an infected donor animal), the airborne transmission efficiency was 71.43% (percentage of all transmission events resulting in an infected sentinel/all transmission events). While IM vaccination reduced airborne transmission to 40% ($N = 10$, $p = 0.3348$, Fisher's exact test, two sided: Odds ratio = 3.75), IN vaccination ($N = 6$, $p = 0.1026$, Fisher's exact test, two sided: Odds ratio = 12.5) reduced it to 16.67% and PI ($N = 2$, $p = 0.1667$, Fisher's exact test, two sided: Odds ratio = not calculable) reduced it to 0% (Fig. 2d). It is possible that we did not see infection in some donor animals, because our sampling scheme was not stringent enough. Therefore, we also compared the airborne transmission efficiency using the data across all transmission events. For naïve hamsters, the airborne transmission efficiency was 63%. While IM vaccination reduced airborne transmission to 29% ($p = 1.870$, Fisher's exact test, two sided: Odds ratio = 4.167), both IN vaccination ($p = 0.0109$, Fisher's exact test, two sided: Odds ratio = 21.67) and PI ($p = 0.0109$, Fisher's exact test, two sided: Odds ratio = 21.67) reduced it to 7%.

Next, we compared the magnitude of overall shedding (AUC of sgRNA recovered in oral swabs on 2, 3, and 5 DPE) between naïve sentinels 1 exposed to naïve donors, and the IM, IN and PI sentinels 1 and their respective naïve controls (Fig. S3a). We combined sentinels across airborne and contact chains. For IM vaccinated, IN vaccinated, and PI sentinels 1, cumulative shedding was significantly reduced compared to naïve sentinels 1 ($p = 0.0379$ (IM vaccinated), $p = 0.0243$ (IN vaccinated), and $p = 0.0040$ (PI), $N = 6$, two-way ANOVA, followed by Šídák's multiple comparisons test). In contrast, while naïve controls shed similar amounts to naïve sentinels 1 in the IM group, we only

observed significant reduction in cumulative shedding in naïve controls in the IN vaccinated group ($p = 0.001$) and the PI group ($p = 0.004$). When excluding all animals with no detectable sgRNA in any oral swab, the magnitude of cumulative shedding did not differ between sentinels with pre-existing immunity and their respective naïve controls. We observed a similar pattern when comparing lung pathology as measured by lung:body weight ratio (Fig. S3b). Comparing sentinels 1, vaccination and previous infection offered significant protection ($p = 0.0432$ (IM vaccinated), $p = 0.033$ (IN vaccinated), and $p = 0.002$ (PI), $N = 6$, two-way ANOVA, followed by Šídák's multiple comparisons test). Protection was also increased for naïve controls, but it was only significant in the PI group ($p = 0.0238$). We did not see a significant difference in the protection from lung pathology between sentinels with pre-existing immunity and their respective naïve controls.

## Pre-existing humoral immunity against Lineage A or Delta B.1.617.2 offers minimal neutralizing cross-reactivity against Omicron B.1.1.529

We inoculated or exposed all animals in these transmission experiments to a mixture of Delta B.1.617.2 and Omicron B.1.1.529. We hypothesized that under pre-existing immune pressure, the competition between Delta B.1.617.2 and Omicron B.1.1.529 would favor Omicron B.1.1.529 due to the larger antigenic distance relative to the previous lineages of SARS-CoV-2. To quantify the immune pressure against Omicron B.1.1.529 in our groups, IgG anti-spike responses were analyzed. All animals seroconverted by day 21 post vaccination or infection with Delta B.1.617.2 (Fig. 3a). Compared to IM vaccination, IN vaccination led to 4-fold higher humoral responses (median titer IM vaccinated = 25,600; median titer IN vaccinated = 102,400, $p = 0.0032$, Kruskal–Wallis test, followed by Dunn's multiple comparison test, $N = 16$). PI hamsters had significantly higher titers (median = 409,600) than both IN vaccinated ($p = 0.0103$) and IM vaccinated ($p < 0.0001$) hamsters. To better assess the production of binding antibodies in the IM, IN, and PI groups, we analyzed the positive sera on a MESO QuickPlex panel[26] (Fig. 3b). In the IM and IN groups, the highest median signal was seen with an antibody response to Lineage A (IM group = 10788.75; IN group = 18692.00; PI group = 81855.75), which supports results from previous studies with vaccines against Lineage A[31]. While the response was strongest against Delta B.1.617.2 in the PI group, the overall response pattern to different variants was similar across all three groups. The median response signal against Omicron B.1.1.529 was consistently lower than against Delta B.1.617.2. Next, a live virus neutralization assay was performed against Delta B.1.617.2 and Omicron B.1.1.529. Neutralizing antibody titers were highest in the PI group, which neutralized Delta B.1.617.2 > 10-fold better than Omicron B.1.1.529 ($p < 0.0001$, $N = 16$, two-way ANOVA followed by Šídák's multiple comparisons test) (Fig. 3c). In the IN vaccinated hamsters, 9 out of 16 animals showed no neutralizing antibodies against the Omicron B.1.1.529 variant. Of IM vaccinated hamsters, 14 out of 16 had no neutralization of the Delta B.1.617.2 variant and 15 out of 16 had no neutralization of the Omicron B.1.1.529 variant. Consequently, virus neutralizing

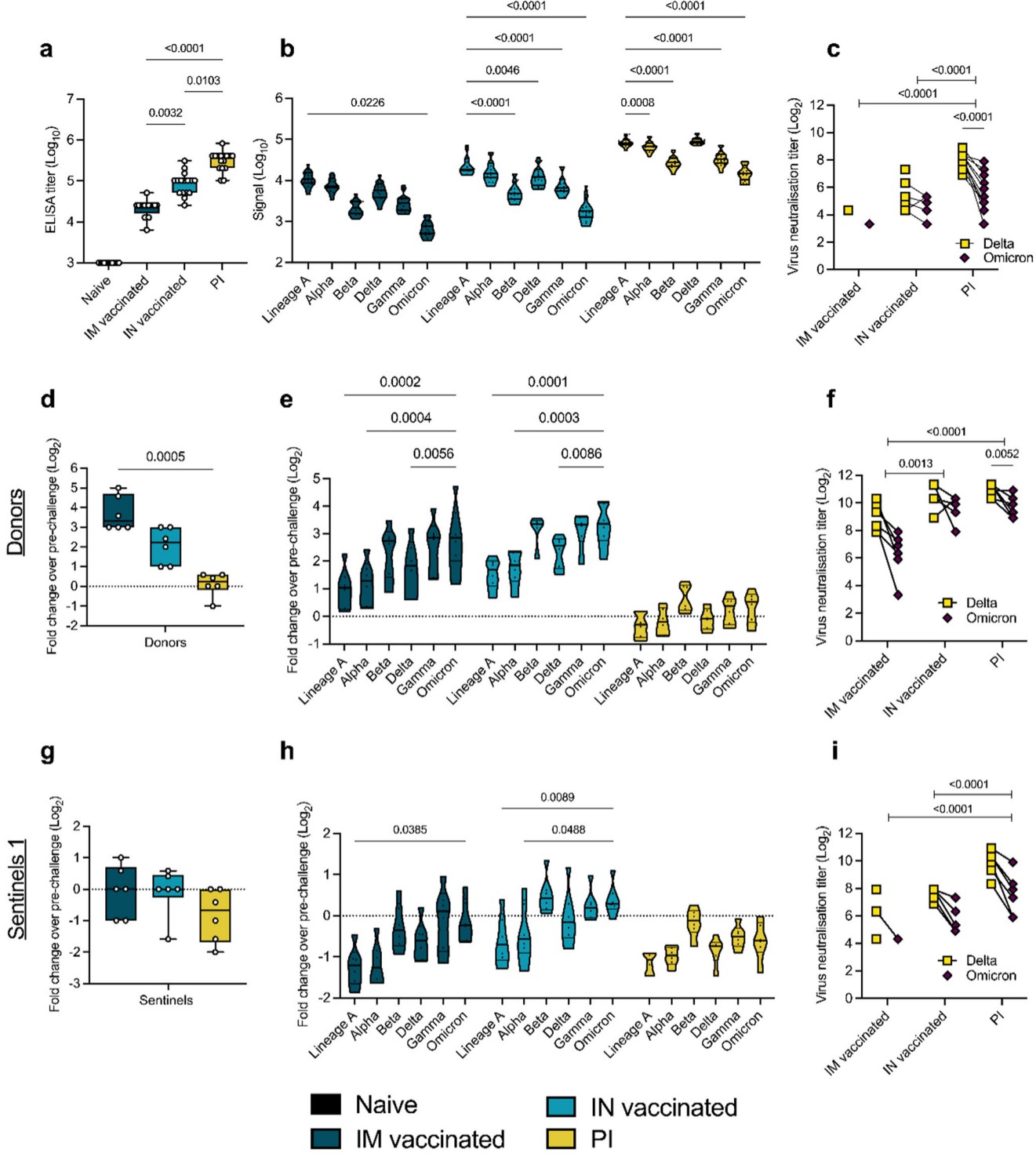

capacity was significantly higher in the PI group as compared to the IM and IN vaccinated groups ($p < 0.0001$, $N = 16$, two-way ANOVA followed by Tukey's multiple comparisons test). These results indicate that prior infection produced the most robust neutralizing antibody response, and that this response is more effective against the Delta B.1.617.2 variant than the Omicron B.1.1.529 variant.

Next, for each donor hamster, the fold change in post-challenge antibody titer relative to their pre-challenge baseline was calculated in samples collected at 5 DPI (Fig. 3d). IM donors experienced a median 10-fold change which was higher as compared to IN donors, which had a median fold-change value of 4.7, and PI donors, which had a titer fold-change of 1.2. These results indicate greater increases in IgG titers in

response to challenge in hamsters that had lower antibody titers at baseline. Variant specific fold-change increase confirmed this finding. Interestingly, the challenge with the 1:1 Omicron B.1.1.529/Delta B.1.617.2 inoculum induced the same affinity maturation profile across groups. The largest fold-change increase was observed for the antigenically most distant variants as compared to the initial priming variant, namely Beta, Gamma, and Omicron B.1.1.529 (Fig. 3e). Intriguingly, the relative difference in fold-change neutralization capacity against Omicron B.1.1.529 compared to Delta B.1.617.2 decreased. Yet, all groups maintained higher levels of neutralizing antibodies against Delta B.1.617.2 than Omicron B.1.1.529, with median titers against Delta B.1.617.2 4-fold higher than Omicron B.1.1.529 in PI animals ($p = 0.0052$,

**Fig. 3 | Variant specific infection- or vaccine mediated humoral immunity.**
Serology pre- and post-challenge with Delta B.1.617.2/Omicron B.1.1.529. Serum was collected 21 days post vaccination against Lineage A or infection with Delta B.1.617.2, and on 5 DPI/DPE. **a** Anti-spike IgG response (Lineage A spike), measured by ELISA. Whisker-plots depicting median, upper and lower quantile, min and max values, and individuals. Kruskal–Wallis test, $N = 16$ animals. **b** Cross-reactivity of the IgG response, measured by Meso QuickPlex. Violin plots depicting median, upper (75th) and lower (25th) quantiles, and individual values. Two-way ANOVA, followed by Šídák's multiple comparisons test. $N = 16$. **c** Individual neutralizing antibody titers against Delta B.1.617.2 and Omicron B.1.1.529. Points connected by lines indicate the same animal. Delta B.1.617.2 and Omicron B.1.1.529: Two-way ANOVA, followed by Šídák's multiple comparisons test. Groups of pre-existing immunity: Two-way ANOVA, followed by Tukey's multiple comparisons test. $N = 16$. **d**, **g** Change in overall anti-spike IgG response after challenge (donors (**d**) and sentinels 1 (**g**)). Whisker-plots depicting median, upper and lower quantile, min and

max values, and individual values. Change in titer is represented as $\text{Log}_2$ (fold change over pre-challenge value). Dotted line indicates no change. Kruskal–Wallis test, $N = 6$. **e**, **h** Change in cross-reactivity after challenge/re-infection. Violin plots depicting median, upper and lower quantiles, and individual values. Change in titer is represented as $\text{Log}_2$ (fold change over pre-challenge value). Dotted line indicates no change in titer. Two-way ANOVA, followed by Šídák's multiple comparisons test. $N = 6$ (donors (**e**) and sentinels 1 (**h**)). **f**, **i** Individual neutralizing antibody titers against Delta B.1.617.2 and Omicron B.1.1.529 after challenge. Points connected by lines indicate the same animal. Delta B.1.617.2 and Omicron B.1.1.529: Two-way ANOVA, followed by Šídák's multiple comparisons test. Groups of pre-existing immunity: Two-way ANOVA, followed by Tukey's multiple comparisons test. $N = 6$ (donors (**f**) and sentinels 1 (=**i**)). black = naïve, dark blue = intramuscularly (IM) vaccinated, light blue = intranasally (IN) vaccinated, yellow = previously infected (PI). P-values stated were significant (<0.05). Source data are provided as a Source Data file.

$N = 6$, two-way ANOVA followed by Šídák's multiple comparisons test) (Fig. 3f). Overall, virus neutralizing capacity was still significantly higher in the PI group as compared to the IM vaccinated, but not the IN vaccinated group ($p < 0.0001$, $N = 6$, two-way ANOVA followed by Tukey's multiple comparisons test). We then investigated the change in antibody profiles in the sentinels 1 group. We observed a positive fold change in post-challenge antibody titer relative to their pre-challenge baseline in only 2 out of 6 sentinels 1 in the IM and IN vaccinated group, and in no PI sentinel 1 (Fig. 3g). In all other sentinels 1, anti-spike antibody levels decreased compared to pre-challenge. This was supported by the variant-specific changes across groups (Fig. 3h), which also revealed profiles like those observed in the donors. We observed a minimal boost in neutralizing capacity across all sentinel 1 groups, which maintained higher levels of neutralizing antibodies against Delta B.1.617.2 than Omicron B.1.1.529, with median titers against Delta B.1.617.2 > 5-fold higher than Omicron B.1.1.529 in PI animals ($p < 0.0001$, $N = 6$, two-way ANOVA followed by Šídák's multiple comparisons test) (Fig. 3i). Raw values for all animals in the transmission chains (donors and sentinels) can be found in Table S3. We next assessed if the strength of the humoral response correlated with the risk of infection upon challenge (donors) or exposure (sentinels 1). In donors, the number of sgRNA positive samples correlated significantly with the magnitude of the anti-spike ELISA titer ($p = 0.0066$, $n = 18$, Spearman correlation) and the neutralization titer ($p = 0.0321$). Neither were found to be significantly correlated in the sentinels 1 group (Fig. S4).

### Pre-existing immunity impacts Omicron B.1.1.529 intra- and inter-host competitiveness

While pseudotype entry for Omicron B.1.1.529 was reduced compared to Delta B.1.617.2 in the hamster, and structural differences exist between the spikes and ACE2s (Fig. S5a–c), no significant differences in oral shedding were observed between variants in naïve hamsters after single infections (Fig. S5d, e). Hence, next we analyzed the relative composition of each of the VOCs in all sgRNA positive samples from the transmission chains (donors and sentinels) by next generation sequencing (NGS). Delta B.1.617.2 outcompeted Omicron B.1.1.529 both within and between hosts in naïve groups (Fig. 4a). Across all sgRNA positive swabs in donors and sentinels, Delta B.1.617.2 comprised >98% of viral sequences, though some individual variation was observed in swabs. The percentage of Delta B.1.617.2 increased with each subsequent naïve transmission chain in swabs (median percentage Delta B.1.617.2 in donors = 98% (99.9–81.8 95% CI); sentinels 1 = 99% (100–84.4 95% CI); sentinels 2 = 99.5% (100–83 95% CI); and sentinels 3 = 99.8% (99.9–99 95% CI). No Omicron B.1.1.529 was detected in lungs or nasal turbinates (Fig. 4c).

In three hamsters with pre-existing immunity, Omicron B.1.1.529 was the dominant variant (Table S2): day 2 swab of one IM vaccinated contact sentinel, days 2 and 3 swabs of one PI donor, and day 2 swab of

a second PI donor. Overall, Delta B.1.617.2 outcompeted Omicron B.1.1.529 in the directly infected donors and the sentinels across all vaccinated and previously infected groups (Fig. 4b). However, compared to the percentage of Omicron B.1.1.529 sequences in swab samples from the naïve animals (<2%), while not significant, Omicron B.1.1.529 was more prevalent in swab samples from hamsters with pre-existing immunity: Donors: IM vaccinated = 2.4%, IN vaccinated = 8.7%, and PI = 40.6%; Sentinels 1: IM vaccinated = 13.4%, IN vaccinated = 6.0%, and PI = 6.9% (Fig. 4d). This trend did not appear in tissue samples, and no Omicron B.1.1.529 was recovered in the nasal turbinates of either IM vaccinated or PI animals, except for one IN donor (18% Omicron B.1.1.529), nor in the lungs of IM vaccinated animals. No sgRNA was recovered from lungs of IN vaccinated or PI animals. These data suggest that immune pressure may be different between physiological compartments within the host, or that in the hamster model the initial relative advantage provided by pre-existing immunity is rapidly lost once infection is established.

## Discussion

The ongoing circulation of SARS-CoV-2 VOCs and vaccinations have created a highly heterogeneous immune landscape in the human population. Household transmission analyses have revealed that vaccinations against SARS-CoV-2 can be effective in reducing transmission not only for SARS-CoV-2 Lineage A, but also VOCs[32]. Fully vaccinated and booster-vaccinated individuals are generally less susceptible to infection compared to unvaccinated individuals[33]. In experimental studies in the Syrian hamster, low heterologous vaccination-induced antibody titers were linked to a reduction in lower respiratory tract pathology and virus replication[34]. However, vaccine induced-SARS-CoV-2 immunity is typically not sterilizing[35], and transmission and virus replication in the upper respiratory tract are still observed after homologous or heterologous challenge in the hamster. In humans, none of the currently licensed vaccines are able to completely block transmission. In particular with Omicron B.1.1.529, vaccine breakthrough and reinfections have frequently been reported. These are likely driven by a combination of waning immunity and antigenic drift[36]. There is a clear need for the development of vaccines with the potential to reduce upper respiratory tract replication and transmission while maintaining their ability to prevent lower respiratory tract disease.

AZD1222 is a replication-incompetent simian adenovirus–vectored vaccine encoding the Lineage A Spike (S) protein of Wuhan-1. Compared to IM vaccination, mucosal vaccination with the ChAdOx1 COVID19 vaccine (AZD1222) has been shown to be more efficient in preventing upper respiratory tract viral replication and shedding, while retaining the potential to prevent disease in pre-clinical models, including the Syrian hamster, ferrets, and rhesus macaques[30,37]. We found here, that while the AZD1222 vaccine was based on the Lineage A S protein, both IM and IN vaccination provided protection from lung

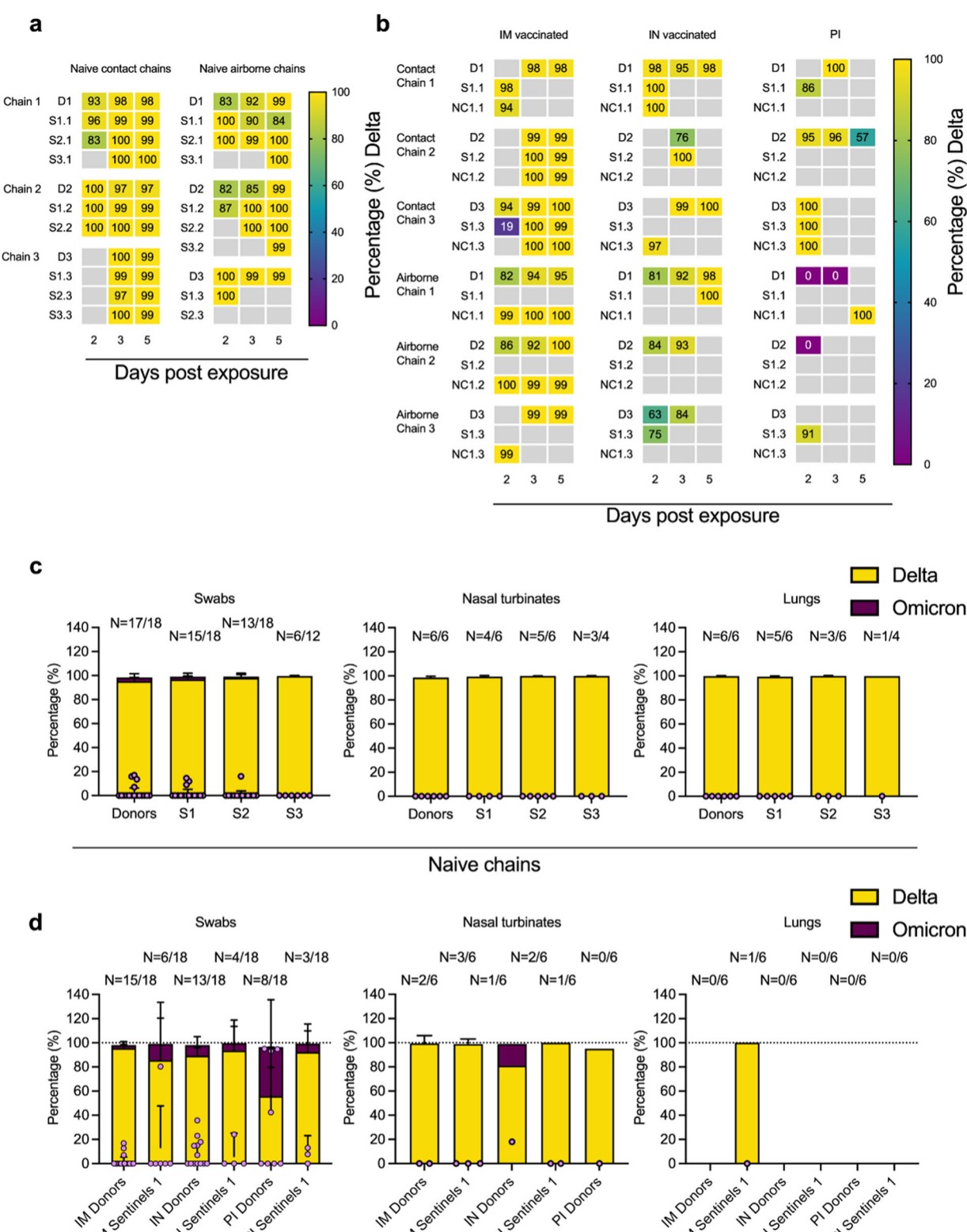

pathology after challenge with a Delta B.1.617.2/Omicron B.1.1.529 mixture in the Syrian hamster. Our data supports increased protection of the lower respiratory tract after IN vaccination against Lineage A or previous infection with Delta B.1.617.2 compared to IM vaccination against Lineage A. As we and others previously demonstrated[30,37], in this study the mucosal vaccination also decreased the viral load in the upper

respiratory tract as compared to IM vaccination. However, only in the PI animals did we observe a significant reduction in cumulative shedding compared to naïve hamsters.

Mucosal COVID-19 vaccines have been experimentally shown to reduce upper respiratory shedding, but not transmission, when assessed in a single contact transmission chain setting[38–40]. In addition,

**Fig. 4 | Competitiveness of Delta B.1.617.2 and Omicron B.1.1.529 in animals naïve to SARS-CoV-2 and with pre-existing immunity. a** The receptor binding domain of the SARS-CoV-2 spike was sequenced for all sgRNA positive swabs collected at 2, 3, and 5 DPI/DPE, and lungs and nasal turbinates collected on 5 DPI/DPE. Heatmap representing all sgRNA positive swab samples from each individual for each naïve chain and showing the percentage of Delta B.1.617.2 detected. Colors refer to legend on right (D = donor, S = sentinel, NC = naïve control), grey = no sgRNA present in the sample or sequencing unsuccessful. **b** Heatmap displaying all sgRNA positive samples from each individual for vaccinated or previously infected chains and showing percentage of Delta B.1.617.2 detected. Colors refer to legend on right (D = donor, S = sentinel, NC = naïve control), grey = no sgRNA present in the sample or sequencing unsuccessful. **c** Overall percentage of Delta B.1.617.2 and Omicron B.1.1.529 in all sgRNA positive samples in each naïve group, separated by tissue type. **d** Overall percentage of Delta B.1.617.2 and Omicron B.1.1.529 in all sgRNA positive samples in each vaccinated or previously infected group, separated by tissue type. Bar charts depicting mean and 95% CI. Number of sgRNA positive samples over all samples analyzed is indicated on top, individual data points are depicted (dots represent percentage of Omicron B.1.1.529, light purple). Yellow = Delta B.1.617.2, purple = Omicron B.1.1.529. Source data are provided as a Source Data file.

hamster studies have shown that previous infection protects against disease, but not upper respiratory tract replication, after homologous and heterologous reinfection[35,41–43]. Similar dynamics were observed in our experimental setup. Vaccination did not completely block the transmission in the first round, but a disruption of the airborne transmission chain was achieved in the second iteration of the transmission chain. Our data show that vaccination resulted in markedly changed transmission and disease dynamics, and this effect was greater for IN than IM vaccination. Vaccination and PI also reduced the magnitude of shedding and disease severity in the lungs in sentinel animals as compared to sentinels exposed to naïve donors. This suggests that pre-existing humoral immunity of the donor not only significantly reduces the likelihood of the first transmission event but also may impact the onwards transmission on a population level, magnifying the effect. Our findings are, due to the small group sizes, observational and additional targeted work could provide statistical confirmation that intranasal vaccination and previous exposure are indeed more capable to block transmission compared to intramuscular vaccination.

The ability to block transmission appears to be related to the strength of the humoral immune response, in which we observe a change in strength from IM to IN to PI. Whereas the antigenic differences between the original ancestral SARS-CoV-2 and most VOCs are relatively minimal, the exception is Omicron B.1.1.529[21]. On average, a drop in neutralizing titers of ~40 has been observed in sera from vaccinated and previously infected individuals[34,44–46].

Although Omicron B.1.1.529 showed reduced transmission potential in the Syrian hamster model, which is a relevant limitation to the work presented here, we confirmed the ability of this VOC to transmit if the exposure window lasted for 24 h[47]. Delta B.1.617.2 out-competed Omicron B.1.1.529 in naïve hamster within and between hosts, suggesting overall greater fitness of Delta B.1.617.2 in that context, even though we confirmed through sequencing that the ratio of genomic material in the inoculum may have favored Omicron B.1.1.529 to begin with. However, in PI donor animals with pre-existing humoral immunity against Delta B.1.617.2, the relative frequency of Omicron B.1.1.529 increased compared to Delta B.1.617.2. Due to the small sample size across groups, especially in the sentinel groups which were protected from transmission, these findings do not provide statistical significance. Drawing definite conclusions is therefore not possible and further investigation is required to understand tissue type-specific effect of pre-existing immunity on viral competitiveness and effects on transmissibility. However, we observed that the gap in neutralizing capacity between Omicron B.1.1.529 and Delta B.1.617.2 decreased more in the IN vaccinated and PI groups after challenge/re-infection with the Delta B.1.617.2/Omicron B.1.1.529 mixture as compared to IM vaccinated animals. This could further suggest that in these groups, a replication advantage was present for Omicron B.1.1.529 initially, which led to increased antibody affinity maturation towards this variant. While we could not report Omicron B.1.1.529 as the dominant variant in most of our animals with pre-existing immunity, our findings align with observations from another study where the authors showed that the presence of neutralizing antibodies against Delta B.1.617.2, but not Omicron B.1.1.529, could prevent Delta B.1.617.2 from outcompeting Omicron B.1.1.529 in hamsters[48]. This suggests, that even in hamsters, where Delta

B.1.617.2 is intrinsically more transmissible, immune pressure can provide a direct advantage for antigenically different viruses.

Our data demonstrate that pre-existing immunity and route of exposure directly influence disease manifestation and onwards transmission efficacy. These data highlight the need to better understand SARS-CoV-2 transmission dynamics amidst the complexity of pre-existing immunity and the emergence of VOCs.

## Methods
### Ethics statement
All animal experiments were conducted in an AAALAC International-accredited facility and were approved by the Rocky Mountain Laboratories Institutional Care and Use Committee following the guidelines put forth in the Guide for the Care and Use of Laboratory Animals 8th edition, the Animal Welfare Act, United States Department of Agriculture and the United States Public Health Service Policy on the Humane Care and Use of Laboratory Animals. Protocol numbers 2021-034-E and 2021-048-E. Work with infectious SARS-CoV-2 virus strains under BSL3 conditions was approved by the Institutional Biosafety Committee (IBC). For the removal of specimens from high containment areas, virus inactivation of all samples was performed according to IBC-approved standard operating procedures.

### Cells and viruses
The SARS-CoV-2 isolates used in this study are summarized in Table S4. Virus propagation was performed in VeroE6 cells (kindly provided by Ralph Baric, University of North Carolina, Chapel Hill, USA; also available as VERO C1008 from ATCC (CRL-1586)) in DMEM supplemented with 2% fetal bovine serum, 1 mM L-glutamine, 50 U/mL penicillin and 50 µg/mL streptomycin (DMEM2). VeroE6 cells were maintained in DMEM supplemented with 10% fetal bovine serum, 1 mM L-glutamine, 50 U/mL penicillin and 50 µg/ml streptomycin. At regular intervals mycoplasma testing was performed. No mycoplasma or contaminants were detected. All virus stocks were sequenced; and no SNPs compared to the patient sample sequence were detected.

### Challenge and transmission studies
**Groups.** Four-to-six-week-old female and male Syrian hamsters (ENVIGO, Hsd Han AURA) were used. Sex was not considered as a variable in this study which did not aim to address sex-differences in vaccine- or previous exposure-induced immunity. Hamsters were randomly assigned to one of four groups: Naïve group, intramuscularly (IM) vaccinated group, intranasally (IN) vaccinated group, and previously infected (PI) group. For the IM vaccinated group, 16 animals received vaccine AZD1222 ($2.5 \times 10^8$ IU/animal) intramuscularly to two sites using a 25-gauge needle with a maximum injection volume of 200 µL. For the IN vaccinated group, 16 animals received vaccine AZD1222 ($2.5 \times 10^8$ IU/animal) intranasally with a maximum injection volume of 60 µL. For the PI group, 16 naïve animals were exposed to Delta B.1.617.2 infected animals in direct contact over multiple days: Four hamsters were inoculated via the intranasal route with a total maximum dose of $10^4$ TCID$_{50}$ SARS-CoV-2 Delta B.1.617.2 VOC. One infected hamster was co-housed with four naïve animals to allow for

contact transmission to occur (ratio 1:4). At least 21 days post vaccination or exposure blood was collected for serology.

**Challenge.** For each group, $N = 6$ animals were infected intranasally with $1 \times 10^4$ TCID$_{50}$ SARS-CoV-2 at a 1:1 ratio of Omicron B.1.1.529 and Delta B.1.617.2 and individually housed. The ratio between Omicron B.1.1.529 and Delta B.1.617.2 was based on TCID$_{50}$ values. The experiment was conducted across three interactions, which each included all groups, due to space constraints using the transmission cages. The inoculum was sequenced by NSG (as described below) and we found 34.2, 35.1, and 34.1 percentage of reads to map to Delta B.1.617.2, respectively. These animals also served as donor animals for the transmission studies described subsequently. Animals were monitored until 5 DPE. Oropharyngeal swabs were taken for all animals at 2, 3, and 5 DPI. All animals were euthanized at 5 DPI for collection of lung tissue and nasal turbinates, and serum.

**Transmission.** The transmission chains were conducted at least 28 days post vaccination or previous infection (infection of donor animals occurred between days 34 and 51, exposure of sentinels occurred between days 35 and 52). Naïve controls were age matched. Naïve group: Donor hamsters ($N = 6$) were infected intranasally and individually housed. After 24 h, three donor animals were placed into a new rodent cage and three donors were placed into the donor cage of an airborne transmission set-up of 16.5 cm distance at an airflow of 30 cage changes/h as described by ref. 24. Sentinels (sentinels 1, $N = 3$) were placed into either the same cage (contact, $N = 3$, 1:1 ratio) or the sentinel cage of the airborne transmission caging (airborne, $N = 3$, 1:1 ratio). Hamsters were co-housed for 48 h. Donor animals were re-housed into regular rodent caging and sentinels 1 were placed into either a new rodent cage or the donor cage of a new airborne transmission set-up. New sentinels (sentinels 2, $N = 3$ for contact and $N = 3$ for airborne) were placed into the same new rodent cage or the sentinel cage of the airborne transmission caging (1:1) at 16.5 cm distance at an airflow of 30 changes/h. Hamsters were co-housed for 48 h. Sentinels 1 were then re-housed into regular rodent caging and 4 sentinels 2 were placed into either a new rodent cage or the donor cage of a new airborne transmission set-up. New sentinels (sentinels 3, $N = 2$ for contact and $N = 2$ for airborne) were placed into the same new rodent cage or the sentinel cage of the airborne transmission caging (1:1) at 16.5 cm distance at an airflow of 30 changes/h. Hamsters were co-housed for 72 h. Then all were re-housed to regular rodent caging and monitored until 5 DPE.

**Vaccinated groups.** Donor hamsters ($N = 6$ each for IM and IN, respectively) were infected intranasally and individually housed. After 24 h, three donor animals were placed into a new rodent cage and three donors were placed into the donor cage of an airborne transmission set-up. Equally vaccinated sentinels (sentinels 1) and completely naïve animals (naïve controls) were placed into either the same cage (contact, $N = 3$, 1:2 ratio) or the sentinel cage of the airborne transmission caging (airborne, $N = 3$, 1:2 ratio).

**PI group.** Donor hamsters ($N = 6$) were infected intranasally and individually housed. After 24 h, three donor animals were placed into a new rodent cage and three donors were placed into the donor cage of an airborne transmission set-up. Equally PI sentinels (sentinels 1) and completely naïve animals (naïve controls) were placed into either the same cage (contact, $N = 3$, 1:2 ratio) or the sentinel cage of the airborne transmission caging (airborne, $N = 3$, 1:2 ratio). Hamsters were co-housed for 48 h.

A second and third chain link was performed for all pre-existing immunity groups for 2/3 airborne links as described above for the naïve chains (sentinel 1 → sentinel 2 (and naïve control 2) → sentinel 3 (and naïve control 3)). Donor animals and sentinels were re-housed into regular rodent caging and monitored until 5 DPE. Oropharyngeal

swabs were taken for all animals at 2, 3, and 5 DPI/DPE. All animals were euthanized at 5 DPI/DPE for collection of lung tissue and nasal turbinates and serum. To ensure no cross-contamination, the donor cages and the sentinel cages were never opened at the same time, sentinel hamsters were not exposed to the same handling equipment as donors, and the equipment was disinfected with either 70% ETOH or 5% Microchem after use. Regular bedding was replaced by alpha-dri bedding to avoid the generation of dust particles.

## Viral RNA detection

Swabs from hamsters were collected as described above. Then, 140 μL was utilized for RNA extraction using the QIAamp Viral RNA Kit (Qiagen) using QIAcube HT automated system (Qiagen) according to the manufacturer's instructions with an elution volume of 150 μL. For tissues, RNA was isolated using the RNeasy Mini kit (Qiagen) according to the manufacturer's instructions and eluted in 60 μL. Sub-genomic (sg) and genomic (g) viral RNA was detected by qRT-PCR[49].

sgRNA: Fw=CGATCTCTTGTAGATCTGTTCTC,
Rv=ATATTGCAGCAGTACGCACACA,
Probe=FAM-ACACTAGCCATCCTTACTGCGCTTCG-ZEN-IBHQ;
gRNA: Fw=AACAGGTACGTTAATAGTTAATAGCGT,
Rv=ATATTGCAGCAGTACGCACACA,
Probe=FAM-ACACTAGCCATCCTTACTGCGCTTCG-ZEN-IBHQ.

RNA was tested with TaqMan™ Fast Virus One-Step Master Mix (Applied Biosystems) using QuantStudio 3 Flex Real-Time PCR System (Applied Biosystems). SARS-CoV-2 standards with known copy numbers were used to construct a standard curve and calculate copy numbers/mL or copy numbers/g. The detection limit for the assay was 10 copies/reaction, and samples below this limit were considered negative.

## ELISA

Serum samples were analyzed as previously described[50]. In brief, maxisorp plates (Nunc) were coated with 50 ng Lineage A spike protein (generated in-house) per well. Plates were incubated overnight at 4 °C. Plates were blocked with casein in phosphate buffered saline (PBS) (ThermoFisher) for 1 h at room temperature. Serum was diluted 2-fold in blocking buffer and samples (duplicate) were incubated for 1 h at room temperature. Secondary goat anti-hamster IgG Fc (Cat.No. 5220-0371 Lot. 10492253, Seracare; each lot is tested to assure specificity and lot-to-lot consistency using an in-house ELISA assay. Reference number: 14-22-06 (https://www.seracare.com/AntiHamster-IgG-HL-Antibody-PeroxidaseLabeled-5220-0371/)) spike-specific antibodies were used (diluted at 2500X) for detection and visualized with KPL TMB 2-component peroxidase substrate kit (SeraCare, 5120-0047). The reaction was stopped with KPL stop solution (Seracare) and plates were read at 450 nm. The threshold for positivity was calculated as the average plus 3 x the standard deviation of negative control hamster sera. Endpoint titers were determined.

## MESO QuickPlex assay

The V-PLEX SARS-CoV-2 Panel 23 (IgG) kit from Meso Scale Discovery was used to test binding antibodies against the spike protein of the different SARS-CoV-2 VOCs, with serum obtained from hamsters 14 DPI diluted at 10,000X. A standard curve of pooled hamster sera positive for SARS-CoV-2 spike protein was serially diluted 4-fold. To prepare a secondary antibody, a goat anti-hamster IgG cross-adsorbed secondary antibody (ThermoFisher, SA5−10284; each lot is tested to assure specificity and lot-to-lot consistency using an in-house assay. Reference number AB_2868332 (https://www.thermofisher.com/antibody/product/Goat-anti-Syrian-Hamster-IgG-H-L-Cross-Adsorbed-Secondary-Antibody-Polyclonal/SA5-10284) was conjugated using the MSD GOLD SULFO-TAG NHS-Ester Conjugation Pack (MSD). The secondary antibody was diluted 10,000X. The plates were prepped, and samples were run according to the kit's instruction manual. After the

plates were read by the MSD instrument, data was analyzed with the MSD Discovery Workbench Application.

## Virus neutralization

Heat-inactivated γ-irradiated sera were two-fold serially diluted in DMEM. 100 $TCID_{50}$ of SARS-CoV-2 were added. After 1 h of incubation at 37 °C and 5% $CO_2$, the virus:serum mixture was added to VeroE6 cells. CPE was scored after 5 days at 37 °C and 5% $CO_2$. The virus neutralization titer was expressed as the reciprocal value of the highest dilution of the serum which still inhibited virus replication.

## Cytokine gene expression

RNA was extracted from hamster lung and nasal turbinates tissue using the RNeasy kit (Qiagen) as per manufacturer's instructions. The expression of five host genes was determined using primer/probe sets derived from ref. [51] and ref. [52] (Table S5). Expression of the following genes was determined: IFNy, TNFa, IL-4, IL-6, and IL-10. Results were normalized to Rpl18 and B2m levels. Fold changes in expression levels were determined using the 2-(Delta) (Delta) Ct method comparing immunized SARS-CoV-2 challenged animals to naïve SARS-CoV-2 challenged animals.

## Next-generation sequencing of virus

Total RNA was extracted from oral swabs, lungs, and nasal turbinates using the Qia Amp Viral kit (Qiagen, Germantown, MD), eluted in EB, and viral Ct values were calculated using qRT-PCR. Subsequently, 11 µL of extracted RNA were used as template in the ARTIC nCoV-2019 sequencing protocol V.1 (Protocols.io - https://www.protocols.io/view/ncov-2019-sequencing-protocol-bbmuik6w) to generate first-strand cDNA. Five microliters were used as template for Q5 HotStart Polymerase PCR (Thermo Fisher Sci, Waltham, MA) together with 10 uM stock of a single primer pair from the ARTIC nCoV-2019 v3 Panel (Integrated DNA Technologies, Belgium); which amplifies the RBD region (nCoV_Spike_76L_alt3: GGGCAAACTGGAAAGATTGCTGA; nCoV_Spike_76R_alt0: ACCTGTGCCTGTTAAACCATTGA). Following 35 cycles and 55 °C annealing temperature, products were AmPure XP cleaned and quantitated with Qubit (Thermo Fisher Sci) fluorometric quantitation as per instructions. Following visual assessment of 1 µL on a Tape Station D1000 (Agilent Technologies, Santa Clara, CA), a total of 400 ng of product was taken directly into TruSeq DNA PCR-Free Library Preparation Guide, Revision D. (Illumina, San Diego, CA) beginning with the Repair Ends step (q.s. to 60 µL with RSB). Subsequent clean-up consisted of a single 1:1 AmPure XP/reaction ratio, and all steps followed the manufacturer's instructions including the Illumina TruSeq CD (96) indexes. Final libraries were visualized on a BioAnalyzer HS chip (Agilent Technologies) and quantified using KAPA Library Quant Kit - Illumina Universal qPCR Mix (Kapa Biosystems, Wilmington, MA) on a CFX96 Real-Time System (BioRad, Hercules, CA). Libraries were diluted to 2 nM stock, pooled together in equimolar concentrations, and sequenced on the Illumina MiSeq instrument (Illumina) as paired-end 2 × 250 base pair reads. Because of the limited diversity of a single-amplicon library, 20% PhiX was added to the final sequencing pool to aid in final sequence quality. Raw fastq reads were trimmed of Illumina adapter sequences using cutadapt version 1.1227, and then trimmed and filtered for quality using the FASTX-Toolkit (Hannon Lab, CSHL). To process the ARTIC data, a custom pipeline was developed[53]. Fastq read pairs were first compared to a database of ARTIC primer pairs to identify read pairs that had correct, matching primers on each end. Once identified, the ARTIC primer sequence was trimmed off. Read pairs that did not have the correct ARTIC primer pairs were discarded. Remaining read pairs were collapsed into one sequence using AdapterRemoval version 2.2.2[54] requiring a minimum 25 base overlap and 300 base minimum length, generating ARTIC amplicon sequences. Identical amplicon sequences were removed, and the unique amplicon sequences were then mapped

to the SARS-CoV-2 genome (MN985325.1) using Bowtie2[55]. Aligned SAM files were converted to BAM format, then sorted and indexed using SAMtools version 1.10[56]. Variant calling was performed using Genome Analysis Toolkit (GATK, version 4.1.2) HaplotypeCaller with ploidy set to 2[57]. Single nucleotide polymorphic variants were filtered for QUAL > 200 and quality by depth (QD) > 20 and indels were filtered for QUAL > 500 and QD > 20 using the filter tool in bcftools, v1.9[56].

## Histopathology

Necropsies and tissue sampling were performed according to IBC-approved protocols. Tissues were fixed for a minimum of 7 days in 10% neutral buffered formalin with 2 changes. Tissues were placed in cassettes and processed with a Sakura VIP-6 Tissue Tek, on a 12-h automated schedule, using a graded series of ethanol, xylene, and PureAffin. Prior to staining, embedded tissues were sectioned at 5 µm and dried overnight at 42 °C. Using GenScript U864YFA140-4/CB2093 NP-1 (1:1000) specific anti-CoV immunoreactivity was detected using the Vector Laboratories ImPress pre-diluted VR anti-rabbit IgG polymer (# MP-6401, (https://vectorlabs.com/products/enzyme-polymer/immpress-vr-horse-anti-rabbit-igg-hrp-kit#documents)) as secondary antibody. The tissues were then processed using the Discovery Ultra automated processor (Ventana Medical Systems) with a ChromoMap DAB kit Roche Tissue Diagnostics (#760-159). Validation of cross-reactivity of SARS-CoV to SARS-CoV-2 in IHC was done in-house by embedding SARS-CoV-2 infected Vero cells in histogel and producing and staining histology slides. Anti-CD3 immunoreactivity was detected utilizing a primary antibody from Roche Tissue Diagnostics predilute (#790-4341, clone 2GV6, validated against control tissues (e.g., spleen) by the manufacturer and in house), secondary antibody from Vector Laboratories ImPress VR horse anti-rabbit IgG polymer (#MP-6401) and visualized using the ChromoMap DAB kit from Roche Tissue Diagnostics (#760-159).

Anti-PAX5 immunoreactivity was detected utilizing a primary antibody from Novus Biologicals at 1:500 (#NBP2-38790, polyclonal, validated by manufacturer, secondary antibody from Vector Laboratories ImPress pre-diluted VR horse anti-rabbit IgG polymer (#MP-6401) and visualized using the ChromoMap DAB kit from Roche Tissue Diagnostics (#760-159). Histopathological assessment was performed by a board-certified, blinded pathologist. Nucleoprotein reactivity was assessed by the pathologist.

## Morphometric analysis

CD3 and PAX5 IHC stained sections were scanned with an Aperio ScanScope XT (Aperio Technologies, Inc., Vista, CA) and analyzed using the ImageScope Positive Pixel Count algorithm (version 9.1). The default parameters of the Positive Pixel Count (hue of 0.1 and width of 0.5) detected antigen adequately.

## Statistics and reproducibility

Power analysis was used to predetermine animal group size to allow statistical significance with 99% confidence intervals for assuming a 5-fold difference in virus replication. Animals were randomly assigned to the experimental groups; investigators were not blinded to allocation during the experiments but were blinded during outcome assessment. Data distribution was assumed to be non-normal and non-parametric test were applied where appropriate. No data or animals were excluded from the analysis. Significance tests were performed as indicated where appropriate using Prism 9 (GraphPad Software). Statistical significance levels were determined as follows: NS, $P > 0.05$; *$P \leq 0.05$; **$P \leq 0.01$; ***$P \leq 0.001$; ****$P \leq 0.0001$.

## Reporting summary

Further information on research design is available in the Nature Portfolio Reporting Summary linked to this article.

## Data availability

The data used in this study are available in the FigShare database under accession code https://doi.org/10.6084/m9.figshare.24061197.v1. The data generated in this study are provided in the Source Data file. The Sequencing data generated in this study have been deposited in the BioProject database under the project ID PRJNA991634 with accession numbers SRR25152370 to SRR25152536. All material requests should be sent to Vincent J. Munster, vincent.munster@nih.gov. Source data are provided with this paper.

## Code availability

No custom computer code or algorithm was used to generate results that are reported in the paper and central to its main claims.

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

## Acknowledgements

We would like to thank Bob Fischer and Shane Gallogly for help with experiments. We thank Tina Thomas, Rebecca Rosenke, and Dan Long for assistance with histology; RMVB animal care staff for taking care of the animals. The following reagent was obtained through: CDC: SARS-CoV-2/human/USA/WA-CDC-WA1/2020, Lineage A. BEI Resources, NIAID, NIH: SARS-CoV-2 variant Alpha (B.1.1.7) (hCoV320 19/England/204820464/2020, EPI_ISL_683466), contributed by Bassam Hallis, and variant Delta B.1.617.2 (B.1.617.2/) (hCoV-19/USA/KY-CDC-2-4242084/2021, EPI_ISL_1823618). Variant Beta (B.1.351) isolate name: hCoV-19/USA/MD-HP01542/2021, EPI_ISL_890360, and variant Gamma (P.1) isolate name: hCoV-19/USA/MD-HP03867/2021, EPI_ISL_1468644, were contributed by Johns Hopkins Bloomberg School of Public Health: Andrew Pekosz.

Variant Omicron B.1.1.529 (B.1.1.529. BA.1) isolate name: hCoV-19/USA/GA-EHC-2811C/2021, EPI_ISL_7171744, was contributed Emory University, Emory Vaccine Center: Mehul Suthar. We thank Andrew Pekosz, Mehul Suthar, Emmie de Wit, Brandi Williamson, Sujatha Rashid, Bassam Hallis, Ranjan Mukul, Kimberly Stemple, Bin Zhou, Natalie Thornburg, Sue Tong, Stacey Ricklefs, Sarah Anzick for gracefully sharing viruses or propagating stocks. This work was supported by the Intramural Research Program of the National Institute of Allergy and Infectious Diseases (NIAID), National Institutes of Health (NIH) (1ZIAAI001179-01). This work was part of NIAID's SARS-CoV-2 Assessment of Viral Evolution (SAVE) Program.

## Author contributions

J.R.P., K.C.Y., and N.v.D. designed the studies. J.R.P., K.C.Y., Z.A.W., J.C.R., T.A.S., V.A.A., J.E.S., M.G.H., and E.H. performed the experiments. J.R.P., K.C.Y., J.C.R., Z.A.W., T.A.S., V.A.A., K.B., C.I.S., and C.M. analyzed results. T.L. and S.C.G. provided materials. R.P.G. generated the schematic visualizations. J.R.P., K.C.Y., and V.J.M. wrote the manuscript. J.R.P., K.C.Y., N.v.D., T.L., S.C.G., and V.J.M. edited the manuscript. All co-authors reviewed the manuscript.

## Funding

## Competing interests

The authors declare no competing interests.

## Additional information

[1]Laboratory of Virology, Division of Intramural Research, National Institute of Allergy and Infectious Diseases, National Institutes of Health, Hamilton, MT, USA. [2]Genomics Research Section, Research Technologies Branch, Division of Intramural Research, National Institute of Allergy and Infectious Diseases, National Institutes of Health, Hamilton, MT, USA. [3]Rocky Mountain Visual and Medical Arts Unit, Research Technologies Branch, Division of Intramural Research, National Institute of Allergy and Infectious Diseases, National Institutes of Health, Hamilton, MT, USA. [4]Rocky Mountain Veterinary Branch, Division of Intramural Research, National Institute of Allergy and Infectious Diseases, National Institutes of Health, Hamilton, MT, USA. [5]The Jenner Institute, Nuffield Department of Medicine, University of Oxford, Oxford, UK. [6]Present address: Chinese Academy of Medical Science Oxford Institute; Oxford Vaccine Group, Department of Paediatrics, University of Oxford, Oxford, UK. [8]These authors contributed equally: Julia R. Port, Claude Kwe Yinda. [8]These authors jointly supervised this work: Neeltje van Doremalen, Vincent J. Munster. ✉e-mail: vincent.munster@nih.gov

