## [Peer Review File · Nature Communications]

Infection- or AZD1222 vaccine-mediated immunity reduces SARS-CoV-2 transmission, but increases competitiveness of Omicron in hamstersREVIEWER COMMENTS

Reviewer #1 (Remarks to the Author):

In this study, Port et al conducted SARS-CoV-2 challenge and transmission experiments in Syrian hamsters to examine how they are impacted by prior immunity. Four groups of hamsters were assessed in both experiments: naïve, intranasally vaccinated with AZD1222, intramuscularly vaccinated with AZD1222, and previously infected with Delta. In the first experiment, these animals were challenged intranasally, with vaccination or previous exposure protecting the animals, as shown by reduced viral load in nasal turbinates and lung, viral shedding, and histopathology. The authors then conducted two transmission experiments with new hamsters encompassing the same groups, examining both contact and airborne transmission. Donor hamsters were infected with a 1:1 mixture of Delta and Omicron, which were then exposed to sentinel hamsters for transmission, and this was repeated for a total of three generations. All naïve hamsters were infected in the contact transmission experiment and the majority were infected in the airborne transmission experiment. In contrast, in either of the vaccinated or the previously infected groups, reduced transmission was observed, with intranasal vaccination and previous infection demonstrating the lowest transmission. The authors then examined the humoral response of these mice pre- and post-challenge with the Delta/Omicron mixture by ELISA, Meso QuickPlex, and live virus neutralization. The previously infected group had the highest titers out of the three groups, with greater titers against Delta than Omicron. After challenge, increases in titer were inversely correlated with their titers prior to challenge. Finally, swabs, nasal turbinates, and lungs taken during the course of the transmission experiments were sequenced by next-generation sequencing and the ratio of Delta to Omicron was examined. Across naïve, vaccinated, and previously infected groups, Delta strongly outcompeted Omicron.

The study is interesting, with the manuscript well-written and the findings are appropriate for this Journal. There have been a limited number of in vivo SARS-CoV-2 transmission experiments conducted and that is a strength of the Munster Lab. An investigation of SARS-CoV-2 transmissibility is of natural importance as the pandemic continues. The sample sizes in some components of the experiments are small (n=2 to 3), but it is understandable given the number of groups and complexity of the experiments. The Omicron variant used here (BA.1) is no longer circulating in the human population, but the takeaways should still be of consequence. One major point to raise is that this work is largely overlapping with a previously published report published last year (ref. 41 in this manuscript), and the results are similar, with the differences being that this current work utilized AZD1222 and examined additional generations beyond the initial passage in the transmission experiment. This reviewer suggests that the inclusion of some additional straightforward experiments, primarily focusing on the latter aspect, would better differentiate this current work from the related report and improve the study, although it is up to the editor's discretion whether this is necessary.

Major comments:

- The methodology of the initial infection experiment shown in Figure 1 seems to not be included in the Methods. The variant that was used should at a minimum be indicated, as the current text simply notes "Animals were inoculated with SARS-CoV-2 ..."
- In the transmission experiment (Figure 2C), there seems to be sentinels that are infected whereas the naïve controls from the same generation are not, and this is seen in all the groups. Considering that the naïve hamsters should lack preexisting immunity and be more susceptible, do the authors have an explanation for this observation?
- Continuing from the above point, Figure 3 only examines the donors – except in Figure 3D, which examines sentinels, although it is not specified which sentinels and this should be clarified (presumably sentinel 1?) – it would be straightforward to examine the humoral response across all of the animals including the sentinels/naïve hamsters from each generation, and this may help explain the above observation.
- In the NGS analysis utilized in Figure 4, is the pipeline searching for exact matches? If there is a mutation that arises, how is it handled? Is there any concern that there may be more or less

mutations that arise in hamsters in either Delta or Omicron over generations?

- There are some points raised in the Discussion of the magnitude of preexisting immunity correlating with the block of transmission, which seems appropriate, but the authors should be careful in the wording as they have only looked at humoral immunity mediated by antibody neutralization and have not examined other aspects of humoral immunity (e.g., effector response, etc.) nor cellular immunity (e.g., T-cell response). It would be meaningful to incorporate some examination of cellular immunity (e.g., ELISPOT) to see if this correlates with the observed transmission events or not, especially as the other related study did not look at this in detail. It would also be helpful to connect these statements back to the data; for example, were the infected animals in groups that did not have 100% infection (e.g., donors in IN vaccination or prior infection) the animals that had the lowest titers? The aggregated raw data for individual animals shown in Table S2 are welcomed and perhaps similar data can be included in the supplement for the antibody binding/neutralization.

Minor comments:

- A scale bar should be added to Figure 1G.
- Line 139 has an extraneous quotation mark.
- Some of the text/methods differ from the descriptions in the figure legends. For example, the Figure 1 legend notes: "Animals were inoculated with SARS-CoV-2 at least 28 days later ..." but the text (Lines 90-91) has a more definitive timeline of "28 days after immunization via vaccination or infection"; the Figure 2 legend notes: "48h post exposure for subsequent groups (sentinels \diamond sentinels)" but the methods notes that hamsters were co-housed for 72 h for the terminal generation (sentinel 2 \diamond sentinel 3). Some proofreading to ensure internal consistency throughout is needed.
- There is some inconsistency with the coloring in Figure 4A; in the Naïve contact chains, Chain 1, D1 (93%) and S2.1 (83%) are teal yet in the Naïve airborne chains, Chain 1, D1 (83%) and S1.1 (84%), etc., are in green. Perhaps the two are using different scales?
- The NGS data should be deposited to a public repository such as the NCBI SRA if it has not been deposited yet.

Reviewer #2 (Remarks to the Author):

Port et al investigated the impact of pre-existing immunity on SARS-CoV-2 replication, pathogenicity, transmission and variant selections in hamsters. They showed that pre-existing immunity reduced viral replication, especially in the lower airways, which coincided with reduced disease, and to a certain extent reduced transmission.

Although understanding how pre-existing immunity shapes the dynamics of viral transmission is interesting, the study suffers from a lack of appropriate design. In addition, the presentation of the results lacks clarity and the interpretation of the data/conclusions is unprecise and proper terminology is not always appropriately used.

Please find below the major concerns:

1. Experimental design. Port et al. induced pre-existing immunity in hamsters by either vaccinating intramuscularly (IM group) or intranasally (IN group) animals with an adeno-based vaccine (AZD1222) which expresses the Wuhan spike or by exposing hamsters to donors infected with the Delta variant (PI group). The authors then compare the impact of these different pre-existing immunities upon challenge with a mix of 1:1 Delta: Omicron. In the whole article, they compare the IM, IN and PI groups as if they had been immunized with the same antigen, which is not the case. Especially, the PI group has been immunized with an antigen that was (partially) matched to the challenge mixture, which is not the case of the two vaccinated groups. As an example, the conclusion drawn line 309-310 is incorrect, as the PI group was immunized against Delta, which was in the virus inoculum.

2. How was the mixture 1:1 between Omicron and Delta performed? Based on TCID50 or on sgRNA levels? Showing NGS data from the inoculum is essential to ensure that animals were indeed inoculated with equal amounts of Omicron and Delta. In addition, performing competition experiments with 1:1 ratio is not very strong, as stochastic differences in initial ratios in the inoculum will easily give the advantage to one variant over another.
3. Omicron does not transmit well in hamsters, therefore it is very difficult to test in hamsters whether Omicron would have a transmission advantage in immunized animals.
4. It is not evident that immunization increased Omicron's competitiveness in hamsters. First of all, the error bars in Figure 4D showing the confidence intervals are very large. No statistics were used to support this conclusion. Second of all, it seems like Omicron was better selected in the donors of the PI groups, certainly because these animals were immunized with Delta, offering an advantage of Omicron over Delta. In animals that were immunized through vaccination with Wuhan, proportions of Omicron in the swabs were not significantly different than in naïve animals. Finally, Omicron's proportion in sentinels of the PI groups did not increase, given that Omicron is intrinsically not very transmissible in hamsters.
5. As far as I understood, viral detection was limited to the detection of sgRNA, not infectious virus. Yet, it is written in the material and methods that the authors performed TCID50 assays and plaque assays in air samples. Which air samples? In addition, how were the thresholds for positivity determined?
6. Numbers of the transmission chains are low, which does not allow statistical support for the observed differences. Example: line 162-163: 1/3 and 2/3 are not statistically different.
7. It is not possible to compare the IM, IN and PI chains, as not all donors were positive in the PI chain. As a result, most sentinel animals also did not get infected, especially in the airborne chain. Therefore comparing the transmission efficiency as shown in Figure 2D is not correct.
8. Looking at 5dpi/5Dpe is early to detect a boost of ab response. For the animals that were exposed, it also depends on when transmission occurred.

Minor comments

1. Line 66: The authors should be careful with the terminology used here. The intrinsic transmissibility of alpha, delta or omicron was not necessarily higher than that of Wuhan-like viruses, but the spread in the population, which is different. The authors should here add references to original scientific articles, rather than to a general review on Omicron.
2. Line 75: what other evolutionary pressures are the authors referring to? What do the authors mean directionality of SARS-CoV-2 evolution? Genetic evolution? Antigenic evolution? The directionality of SARS-CoV-2 antigenic evolution will be only influenced by pre-existing immunity.
3. Line 87: indicate here what vaccine strain is in AZD1222
4. Line 87: change "N=6 hamsters per group", to "Six hamsters per group" at the beginning of the sentence.
5. Figure 1: statistical tests described in the text and in the figure legend are not the same. Precise in the legend that A shows data from the lungs and B from the nasal turbinates.
6. Line 93: On viral RNA was detected, which does not allow to conclude about viral replication.
7. Line 114: change "respectably" for "respectively"
8. Figure 1G: adjusting the white balance would help increasing the contrast and readability of the stainings.
9. Figure 1E: how was the NP reactivity quantified?
10. Figure 1D: describe the box plots as in other sub-legends
11. Line 717: "measured" instead of "measure"
12. Figure 1: spell out the abbreviations in the legend
13. Line 129: "To establish the ability and limitations of the naïve Syrian hamster to model transmission", hamsters do not model transmission, but the hamster transmission model.
14. Figure 1A is not very informative. It is difficult to understand how the transmission chains were performed. Perhaps a time line would be easier to understand the different chains of transmission.
15. Line 145: add a coma after SARS-CoV-2
16. Line 152: "If animal was considered infected if 2 out of 5 samples (either a swab, nasal turbinates, or lung tissue sample) had detectable sgRNA, all IM vaccinated donors became infected." Change to "Animals were considered infected if 2 out of 5 samples (either a swab, nasal turbinates, or lung tissue samples) had detectable sgRNA. All IM vaccinated donors became infected"
17. Figure 2B: The scheme is unclear. Although it is indicated in the legend that the colors are described on the right of the figure, it is not the case. In the previously infected group, not clear why

the yellow animals (which were the donors in the infection) were then the donors in the subsequent contact/airborne transmission, rather than the kaki ones.

18. line 162: authors refer to the chain of transmission but only describe the first transmission round. It is confusing.

19. Line 166-167. The sentence is confusing. "the increased reduction in airborne transmission already observed between donors and sentinels 1? The virus did not transmit to the donors.

20. Line 756: Animals were not vaccinated with Delta, but with Wuhan.

21. Figure 3A: which spike was used to detect IgG ab?

22. Figure 3B: which quantiles are indicated?

23. Figure 3C: not clear why there are so few points in the IM group.

24. Figure 3D: are all sentinels animals included here, also those which did not get infected?

25. Figure 4A: not clear what represent the three squares, different days? Although in the legend it is indicated that percentages in the tissues are also shown. In some occasions, there were two animals in chain 2 and in some occasion two animals in chain 3, it is unclear why this was the case.

26. Figure 4B: what mean the C and O in the purple squares? It is not explained in the legend.

27. Figure 4C and 4D: error bars are sometimes up and sometimes down.

Reviewer #3 (Remarks to the Author):

Comments on "Infection- or AZD1222 vaccine mediated immunity reduces SARS-Cov-2 transmission, but increases competitiveness of Omicron in hamsters" (NCOMMS-23-02914)

The authors evaluated the competitive transmission dynamics of Delta and Omicron in Syrian hamsters that were previously (1) immunized intramuscularly (IM) with AZD1222, (2) immunized intranasally (IN) with AZD1222, or (3) recovered from a prior infection (PI) by Delta. Experiments were conducted in parallel with naïve hamster controls. Overall, this is a valuable study and the results are generally sound.

Specific comments:

1. Figure 2D. The denominator for IM, IN and PI groups contained both naïve hamsters and immuned hamsters. I wonder if you can calculate the infection rate separately? If focusing on immuned hamsters, do you still see a significant difference between the groups (lines 177-181).

2. According to Figure 3, PI hamsters showed higher neutralizing antibody response than IN or IM vaccinated hamsters. Was the difference significant? Would you comment on the dose of AZD1222 that was used to vaccinate hamsters? How many days (provide a range if possible) post-vaccination or post-infection were the hamsters exposed to the donors?

We thank the reviewers for their time and useful suggestions which we have addressed in a point-by-point manner in this rebuttal and in the revised manuscript.

Reviewer #1

In this study, Port et al conducted SARS-CoV-2 challenge and transmission experiments in Syrian hamsters to examine how they are impacted by prior immunity. Four groups of hamsters were assessed in both experiments: naïve, intranasally vaccinated with AZD1222, intramuscularly vaccinated with AZD1222, and previously infected with Delta. In the first experiment, these animals were challenged intranasally, with vaccination or previous exposure protecting the animals, as shown by reduced viral load in nasal turbinates and lung, viral shedding, and histopathology. The authors then conducted two transmission experiments with new hamsters encompassing the same groups, examining both contact and airborne transmission. Donor hamsters were infected with a 1:1 mixture of Delta and Omicron, which were then exposed to sentinel hamsters for transmission, and this was repeated for a total of three generations. All naïve hamsters were infected in the contact transmission experiment and the majority were infected in the airborne transmission experiment. In contrast, in either of the vaccinated or the previously infected groups, reduced transmission was observed, with intranasal vaccination and previous infection demonstrating the lowest transmission. The authors then examined the humoral response of these mice pre- and post-challenge with the Delta/Omicron mixture by ELISA, Meso QuickPlex, and live virus neutralization. The previously infected group had the highest titers out of the three groups, with greater titers against Delta than Omicron. After challenge, increases in titer were inversely correlated with their titers prior to challenge. Finally, swabs, nasal turbinates, and lungs taken during the course of the transmission experiments were sequenced by next-generation sequencing and the ratio of Delta to Omicron was examined. Across naïve, vaccinated, and previously infected groups, Delta strongly outcompeted Omicron.

The study is interesting, with the manuscript well-written and the findings are appropriate for this Journal. There have been a limited number of in vivo SARS-CoV-2 transmission experiments conducted and that is a strength of the Munster Lab. An investigation of SARS-CoV-2 transmissibility is of natural importance as the pandemic continues.

We thank the reviewer for their comments and very helpful insights into how to improve this manuscript. We appreciate their attention detail and thoroughness and hope we have addressed all concerns sufficiently.

The sample sizes in some components of the experiments are small (n=2 to 3), but it is understandable given the number of groups and complexity of the experiments. The Omicron variant used here (BA.1) is no longer circulating in the human population, but the takeaways should still be of consequence. One major point to raise is that this work is largely overlapping with a previously published report published last year (ref. 41 in this manuscript), and the results are similar, with the differences being that this current work utilized AZD1222 and examined additional generations beyond the initial passage in the transmission experiment. This reviewer suggests that the inclusion of some additional straightforward experiments, primarily focusing on the latter aspect, would better differentiate this current work from the related report and improve the study, although it is up to the editor's discretion whether this is necessary.

We agree with the reviewer. Unfortunately, these transmission chain experiments are labor-intensive, and we have observed that transmission is reduced after the first generation of

transmission events due to the protection offered by vaccination. This indicates, that if we wanted to conduct further studies focused on the latter aspect, the animal numbers would be become rather large, and this is currently beyond the scope of the manuscript. For example: If we assume based on our data, that we would want to find statistical significance between transmission efficiencies if IN and IM vaccinated groups between sentinels 1 and sentinels 2, we would have to compare e.g., 0% transmission with 30% transmission, this requires 23 animals per group with an alpha = 0.05 and power = 80. If we then take into consideration that we need approx. 3x as many sentinels 1 to ensure we have 23 infected, we are looking at 69 donors, 69 sentinels 1, and 23 sentinels 2 per group.

Major comments:

- The methodology of the initial infection experiment shown in Figure 1 seems to not be included in the Methods. The variant that was used should at a minimum be indicated, as the current text simply notes “Animals were inoculated with SARS-CoV-2 ...”

The reviewer is correct, that this was not clearly stated. In fact, the animals used for this challenge experiment served also as the donor animals for the transmission study. We have clarified this in the methods under the section Line 391: “Challenge and transmission studies”.

- In the transmission experiment (Figure 2C), there seems to be sentinels that are infected whereas the naïve controls from the same generation are not, and this is seen in all the groups. Considering that the naïve hamsters should lack preexisting immunity and be more susceptible, do the authors have an explanation for this observation?

The reviewer is correct in their observation, and we have also noted this. At this stage, we do not have an explanation of this observation beyond the hypothesis, that we could be observing stochastic events. This is likely due to the relatively low amount of infectious virus transmitted during these events.

(Port JR, Morris DH, Riopelle JC, et al. Host and viral determinants of airborne transmission of SARS-CoV-2 in the Syrian hamster. Preprint. bioRxiv. 2023;2022.08.15.504010. Published 2023 Feb 21. doi:10.1101/2022.08.15.504010)

We have now added serology data for the naïve sentinels post-challenge (see new Supplemental Table 3). Even after exposure in the transmission study, the majority of these animals have signal magnitudes significantly below those observed in the pre-existing immunity groups, suggesting that these animals were immunologically naïve pre-challenge.

- Continuing from the above point, Figure 3 only examines the donors – except in Figure 3D, which examines sentinels, although it is not specified which sentinels and this should be clarified (presumably sentinel 1?) – it would be straightforward to examine the humoral response across all of the animals including the sentinels/naïve hamsters from each generation, and this may help explain the above observation.

We have clarified that the previously shown data for the anti-spike ELISA is for the sentinels 1 group. We have now added new panels to the Figure 4 (E – G) to show the anti-spike ELISA, the variant-specific response, and neutralization for the sentinels 1 group. We have also added a new Supplemental Table 3 to show raw values in addition to the fold-change depicted in the figure including the ELISA, virus neutralization and variant-specific responses across donors, sentinels and naïve animals. We suggest that the variant-specific analysis of the naïve animals will be sufficient and shows that most naïve controls do not show reactivity, and that additional neutralization assays on this group will not add additional information.

New section in the results Line 255: Overall, virus neutralizing capacity was still significantly higher in the PI group as compared to the IM, but not the IN, vaccinated group ($p < 0.0001$, $N = 6$, two-way ANOVA followed by Tukey's multiple comparisons test). We then investigated the change in antibody profiles in the sentinels 1 group. We observed a positive fold change in post-challenge antibody titer relative to their pre-challenge baseline in only 2 out of 6 sentinels 1 in the IM and IN vaccinated group, and in no PI sentinel 1 (Figure 3 G). In all other sentinels 1, anti-spike antibody levels decreased compared to pre-challenge. This was supported by the variant-specific changes across groups (Figure 3 H), which also revealed profiles reminiscent of those observed in the donors. We observed a minimal boost in neutralizing capacity across all sentinel 1 groups, which maintained higher levels of neutralizing antibodies against Delta than Omicron, with median titers against Delta >5-fold higher than Omicron in PI animals ($p < 0.0001$, $N = 6$, two-way ANOVA followed by Šídák's multiple comparisons test) (Figure 3 I). Raw values for all animals in the transmission chains (donors and sentinels) can be found in Table S3.

Line 829: G. Change in overall anti-spike IgG response after challenge (sentinels 1). Whisker-plots depicting median, min and max values, and individual values. Change in titer is represented as Log₂ (fold change over pre-challenge value). Dotted line indicates no change in titer. Kruskal-Wallis test, $N = 6$. H. Change in cross-reactivity after challenge/re-infection in sentinels 1. Violin plots depicting median, upper and lower quantiles, and individual values. Change in titer is represented as Log₂ (fold change over pre-challenge value). Dotted line indicates no change in titer. Two-way ANOVA, followed by Šídák's multiple comparisons test. $N = 16$. I. Individual neutralizing antibody titers of sentinels 1 against Delta and Omicron after challenge. Points connected by lines indicate the same animal. To assess differences between Delta and Omicron: Two-way ANOVA, followed by Šídák's multiple comparisons test. To assess differences between groups of pre-existing immunity: Two-way ANOVA, followed by Tukey's multiple comparisons test. $N = 6$. black = naïve, dark blue = IM vaccinated, light blue = IN vaccinated, yellow = previously infected. P-values stated were significant (<0.05).

- In the NGS analysis utilized in Figure 4, is the pipeline searching for exact matches? If there is a mutation that arises, how is it handled? Is there any concern that there may be more or less mutations that arise in hamsters in either Delta or Omicron over generations?

We used a single primer pair from the ARTIC nCoV-2019 v3 Panel (Integrated DNA Technologies, Belgium) to amplify the RBD with sufficient nucleotide differences between Delta

and Omicron variants. Each amplicon was compared to the appropriate references and assigned as Delta or Omicron based on exact matches. Since we did not perform full genome sequences, we could not explore the mutation possibilities of these variants in hamsters. We have modified the methods section to make this clearer (**Line 508**: Five microliters were used as template for Q5 HotStart Polymerase PCR (Thermo Fisher Sci, Waltham, MA) together with 10 uM stock of a single primer pair from the ARTIC nCoV-2019 v3 Panel (Integrated DNA Technologies, Belgium); which amplifies the RBD region.). We did detect SNPs in a small subset of samples, predominantly as minority populations (<25%) except for two samples which have a mutation at position 22920 (A →C). The A to C substitution at position 483 of spike resulted in a silent mutation ASN / ASN.

Sample code	Description	% reads	Position	TYPE	REF	ALT	
NGS_0525_run144_omicron_results							
22759	Nasal_turbinates_CoV1523_Ct-Undet_plt6_C03	15%	22824				
22772	Nasal_turbinates_CoV1536_Ct-Undet_plt6_D04	52%	22920	SNP	A	C	synonymous
22813	Nasal_turbinates_CoV1577_Med_plt6_G09	12%	22930				
22806	Nasal_turbinates_CoV1570_Med_plt6_G02	13%	22937				
NGS_0525_run142_omicron_results							
22315	day2swabs_CoV1522_High_plt1_B02	14%	22855				
22394	tube_day3swabs_CoV1564_Low_plt1_H09	8%	22898				
22591	day5swab_CoV1551_High_plt4_D03	16%	22926				
NGS_0525_run143_omicron_results							
22549	day3swabs_CoV1572_Med_plt3_H04	16%	22750				
22401	day5swab_CoV1521_Med_plt2_A04	7%	22822				
22442	day5swab_CoV1546_Med_plt2_F05	19%	22823				
22442	day5swab_CoV1546_Med_plt2_F05	24%	22824				
22488	day2swabs_CoV1527_High_plt3_C03	100%	22920	SNP	A	C	synonymous
22537	day3swabs_CoV1548_High_plt3_G04	22%	22990				
NGS_0525_run146_omicron_results							
22639	lung_CoV1499_Low_plt5_A03	9%	22750				
22640	lung_CoV1500_Low_plt5_A04	11%	22842				

- There are some points raised in the Discussion of the magnitude of preexisting immunity correlating with the block of transmission, which seems appropriate, but the authors should be careful in the wording as they have only looked at humoral immunity mediated by antibody neutralization and have not examined other aspects of humoral immunity (e.g., effector response, etc.) nor cellular immunity (e.g., T-cell response). It would be meaningful to incorporate some examination of cellular immunity (e.g., ELISPOT) to see if this correlates with the observed transmission events or not, especially as the other related study did not look at this in detail. It would also be helpful to connect these statements back to the data; for example, were the infected animals in groups that did not have 100% infection (e.g., donors in IN vaccination or prior infection) the animals that had the lowest titers? The aggregated raw data

for individual animals shown in Table S2 are welcomed and perhaps similar data can be included in the supplement for the antibody binding/neutralization.

This is an important comment and we have added additional data on the humoral response for the sentinel animals (Revised Figure 3 and Supplemental Table 3). Unfortunately, no samples were collected which could be used to investigate the cellular response in the form suggested by the reviewer. We have modified the language in the discussion to reflect that we only looked at humoral data (Line 338 and 353). Unfortunately, we did not collect any samples which could be used to look at T cell responses in these animals. We were able to generate qRT PCR data from nasal turbinates and lungs and investigated key T cell cytokines in all challenged donor animals on day 5 post challenge. However, due to there not being uninfected animals in this study (which are pre-existing immunity matched), this data is challenging to interpret. Data is depicted as fold-change over the naïve group. Therefore, a negative value suggests a downregulated expression of the respective cytokine gene compared to the naïve challenged group. Interestingly, we find that the responses in the nasal turbinates and the lungs do not align across the investigated cytokines and that the responses are different between pre-existing immunity groups. While this data is not ideal to address the reviewer's comment, we decided to include it in the supplement to increase the reader's understanding of the underlying immune responses responsible for the transmission blockage and reduced pathology.

Line 55: Fig. S2. Changes in cytokine gene expression post-challenge. Nasal turbinate (NT) and lung samples were collected at 5 DPI and the fold-change in mRNA expression was calculated for challenged animals with pre-existing immunity over challenged naïve animals.

Violin plots depicting median, quantiles, and individual values, N = 6. Dark blue = IM vaccinated, light blue = IN vaccinated, yellow = PI. Kruskal-Wallis followed by Mann Whitney if statistically significant; p-values stated were significant (<0.05).

We have added the following statement to the results: **Line 127:** This was accompanied by decreased gene expression levels for INF γ and IL-10 in both the upper and lower respiratory tract, and IL-6 in the lower but not the upper respiratory tract, as compared to naïve animals. Expression levels of TNF α remained unchanged. A trend towards increased IL-4 expression was observed in the upper respiratory tract, especially in PI animals, as compared to naïve controls (Figure S2).

And to the methods: **Line 496:** Cytokine gene expression. RNA was extracted from hamster lung and NT tissue using the RNeasy kit (Qiagen) as per manufacturer's instructions. The expression of five host genes was determined using primer/probe sets derived from 51 and 52. Expression of the following genes was determined: IFN γ , TNF α , IL-4, IL-6, and IL-10. Results were normalized to Rpl18 and B2m levels. Fold changes in expression levels were determined using the 2^{-($\Delta\Delta$ Ct)} method comparing immunized SARS-CoV-2 challenged animals to naïve SARS-CoV-2 challenged animals.

The suggestion of the reviewer to investigate if we observe a correlation between infection and strength of the humoral response has strong merit. We correlated the strength of the pre-challenge antibody response (anti-Delta neutralization and anti-spike ELISA) with the amount of positive PCR samples (3 swab samples and 2 tissue samples) after challenge or exposure. We thought it would be best to separate between donors and sentinel 1 animals and found stronger correlations with the ELISA titer than with the neutralization titer and have added this as a supplemental figure to the manuscript.

Line 81: Fig. S4. Correlation between humoral immune response and protection from infection. Serum was collected at least 21 days post vaccination against Lineage A or infection with Delta. Correlation between anti-spike IgG response (Lineage A spike), measured by ELISA or individual neutralizing antibody titers against Delta and the amount of positive sgRNA swab or tissue samples (>10 copies/rxn) after challenge (donors) or exposure (sentinels 1). Individuals are depicted, as well as a linear regression line. N = 18, Spearman's correlation, p -values indicated.

In the results **Line 268:** We next assessed if the strength of the humoral response correlated with the risk of infection upon challenge (donors) or exposure (sentinels 1). The number of

sgRNA positive samples (>10 copies/rxn) correlated significantly ($p = 0.0066$, $N = 18$, Spearman) with the magnitude of the anti-spike ELISA titer and with the neutralization titer ($p = 0.0321$) for donors. Neither were found to be significantly correlated in the sentinels 1 group (Figure S4).

Minor comments:

- A scale bar should be added to Figure 1G. This has been added.
- Line 139 has an extraneous quotation mark. This has been removed.
- Some of the text/methods differ from the descriptions in the figure legends. For example, the Figure 1 legend notes: “Animals were inoculated with SARS-CoV-2 at least 28 days later ...” but the text (Lines 90-91) has a more definitive timeline of “28 days after immunization via vaccination or infection”; the Figure 2 legend notes: “48h post exposure for subsequent groups (sentinels \diamond sentinels)” but the methods notes that hamsters were co-housed for 72 h for the terminal generation (sentinel 2 \diamond sentinel 3). Some proofreading to ensure internal consistency throughout is needed.

Line 90 has been changed to reflect to what is stated in the figure legend. We thank the reviewer for their attention to details. The legend for figure 2 is correct. It describes when exposure was started, while the methods section refers to the duration. We value the attention to detail and have verified internal consistency throughout the manuscript.

- There is some inconsistency with the coloring in Figure 4A; in the Naïve contact chains, Chain 1, D1 (93%) and S2.1 (83%) are teal yet in the Naïve airborne chains, Chain 1, D1 (83%) and S1.1 (84%), etc., are in green. Perhaps the two are using different scales? We thank the reviewer for catching this. There was indeed a different scale applied to one figure and we have now matched them all to each other across A and B.
- The NGS data should be deposited to a public repository such as the NCBI SRA if it has not been deposited yet.

We have only sequenced the RBD region (amplicon) of samples tested positive by qRT PCR. The data will become available under an accession number upon acceptance of the manuscript.

We have also, to assist with data comprehension, updated Sup. Table 2 to reflect that only sgRNA positive (>10 copies/rxn) were included in the sequencing analysis.

Line 120: Table S2: Shedding and tissue titers for each transmission chain (donors and sentinels 1). IM = intramuscularly vaccinated, IN = intranasally vaccinated, PI = Previously infected, BDL = Below qRT-PCR detection limit, sgRNA = sub-genomic RNA. Swab days = 2, 3 and 5. Samples with < 10 sgRNA copies/rxn (approx. $> ct = 36$) were excluded from sequencing analysis. No data provided if sequencing did not pass quality control or produced no results.

Reviewer #2

Please find below the major concerns:

1. Experimental design. Port et al. induced pre-existing immunity in hamsters by either vaccinating intramuscularly (IM group) or intranasally (IN group) animals with an adeno-based vaccine (AZD1222) which expresses the Wuhan spike or by exposing hamsters to donors infected with the Delta variant (PI group). The authors then compare the impact of these different pre-existing immunities upon challenge with a mix of 1:1 Delta: Omicron. In the whole article, they compare the IM, IN and PI groups as if they had been immunized with the same

antigen, which is not the case. Especially, the PI group has been immunized with an antigen that was (partially) matched to the challenge mixture, which is not the case of the two vaccinated groups. As an example, the conclusion drawn line 309-310 is incorrect, as the PI group was immunized against Delta, which was in the virus inoculum.

We thank the reviewer for their effort and the obvious dedication to improving this work. We hope that we have addressed the comments sufficiently.

We have chosen specifically to use vaccination against Lineage A and exposure to Delta to reflect the immune landscape when Omicron emerged. Based on antigenic cartography, these two variants are rather similar (van Doremalen, N., Schulz, J.E., Adney, D.R. *et al.* ChAdOx1 nCoV-19 (AZD1222) or nCoV-19-Beta (AZD2816) protect Syrian hamsters against Beta Delta and Omicron variants. *Nat Commun* **13**, 4610 (2022). <https://doi.org/10.1038/s41467-022-32248-6>), and cross-neutralization is strong. In this line, based on our variant-specific data (MesoQuick Plex), it appears that even if animals are vaccinated against Lineage A, they still mount a strong response against Delta, much more so than against Omicron. As such, we do respectfully suggest, that there is also immune pressure in favor for Omicron and against Delta in these groups.

To ensure this is clear also to the reader, we have added this reference to the manuscript: **Line 86:** Pre-existing immunity was achieved by intranasal (IN) or intramuscular (IM) vaccination with AZD1222 (against Lineage A), or previous infection with the antigenically close Delta 26

We agree that this is not specific enough and have made sure that throughout the manuscript it remains clear that the vaccination was against Lineage A and the previous exposure against Delta. We have also modified the conclusion in **Line 324, Line 45 and 355** to reflect this.

2. How was the mixture 1:1 between Omicron and Delta performed? Based on TCID₅₀ or on sgRNA levels? Showing NGS data from the inoculum is essential to ensure that animals were indeed inoculated with equal amounts of Omicron and Delta. In addition, performing competition experiments with 1:1 ratio is not very strong, as stochastic differences in initial ratios in the inoculum will easily give the advantage to one variant over another.

The ratio was based on TCID₅₀. We have now analyzed the inoculums with NGS and have added the following to the methods section **Line 407:** The ratio between Omicron and Delta was based on TCID₅₀ values. The experiment was conducted across three interactions, which each included all groups (naïve, IM vaccinated, IN vaccinated and PI), due to space constraints using the transmission cages. The inoculum was sequenced by NSG (as described below) and we found 34.2, 35.1, and 34.1 percentage of reads to map to Delta, respectively.

The reviewer was correct in their assumption, that the TCID₅₀ ratio may not reflect sequencing results. However, it is possible that the discrepancy to the TCID₅₀ result is due the DI particles or a skewed genomic RNA to infectious particle ratio which could differ between the stocks (we have observed difference in TCID₅₀/genomic RNA ratios between different variants of concern). However, to address the reviewer's concern, we have also added a comment to the discussion: **Line 353:** Delta out-competed Omicron in naïve hamster within and between hosts, suggesting overall greater fitness of Delta in that context, even though we confirmed through sequencing that the ratio of genomic material in the inoculum may have favored Omicron to begin with.

While we agree that multiple ratios would be better, this is much easier done in vitro than in an in vivo animal experiment which is work intensive and restricted in animal numbers. As such, we respectfully disagree that additional ratios are required at this stage. We are also utilizing this

study to look at potential population effects, which may out way individual nuances found by using different ratios in vitro.

3. Omicron does not transmit well in hamsters, therefore it is very difficult to test in hamsters whether Omicron would have a transmission advantage in immunized animals.

This is correct and we have already stated in the discussion that:

Line 350: Although Omicron showed reduced transmission potential in the Syrian hamster model, which is a relevant limitation to the work presented here, we confirmed the ability of this VOC to transmit if the exposure window lasted for 24 h⁴².

Line 367: This suggests, that even in hamsters, where Delta is intrinsically more transmissible, immune pressure can provide a direct advantage for antigenically different viruses.

4. It is not evident that immunization increased Omicron's competitiveness in hamsters. First of all, the error bars in Figure 4D showing the confidence intervals are very large. No statistics were used to support this conclusion. Second of all, it seems like Omicron was better selected in the donors of the PI groups, certainly because these animals were immunized with Delta, offering an advantage of Omicron over Delta. In animals that were immunized through vaccination with Wuhan, proportions of Omicron in the swabs were not significantly different than in naïve animals. Finally, Omicron's proportion in sentinels of the PI groups did not increase, given that Omicron is intrinsically not very transmissible in hamsters.

In response to this critical and important comment we have made sure to tone down any suggestion in the manuscript that vaccination (especially through the IM route) impacted Omicron competitiveness significantly. As the reviewer notes, it was most obviously increased in donors of the PI group. However, the antigenic difference between Delta and Lineage A is small similar (van Doremalen, N., Schulz, J.E., Adney, D.R. *et al.* ChAdOx1 nCoV-19 (AZD1222) or nCoV-19-Beta (AZD2816) protect Syrian hamsters against Beta Delta and Omicron variants. *Nat Commun* **13**, 4610 (2022). <https://doi.org/10.1038/s41467-022-32248-6>), and vaccination against Lineage A will provide significant protection against Delta as previously published and shown in our variant specific analysis and neutralization data.

It is difficult to address the competitive advantage in the sentinel group, because nearly no sentinel in the PI group got infected. We acknowledge this and have now added additional wording in the discussion to address this caveat and the caveat, that statistical conclusions are not possible, and we only observe trends.

Discussion **Line 357:** Due to the unfortunately small sample size across groups, especially in the sentinels groups which were protected from transmission, these findings do not provide statistical significance. Drawing definite conclusions is therefore not possible and further investigation is required to understand tissue type specific effect of pre-existing immunity on viral competitiveness and effects on transmissibility. However, we observed...

We have also added individual data points to **Figure 4** to more clearly show how many samples taken from each group had Omicron percentage above those seen in the naïve group.

5. As far as I understood, viral detection was limited to the detection of sgRNA, not infectious virus. Yet, it is written in the material and methods that the authors performed TCID50 assays and plaque assays in air samples. Which air samples? In addition, how were the thresholds for positivity determined?

This is correct and has been edited. No air samples were collected, and all data is sgRNA. We apologize for the confusion this caused. These sections have been removed.

6. Numbers of the transmission chains are low, which does not allow statistical support for the observed differences. Example: line 162-163: 1/3 and 2/3 are not statistically different.

This is correct, and we have added a sentence in the discussion, that future studies should address this in Line 341: Due to the small group sizes our findings are, observational and additional targeted work could provide statistical confirmation that intranasal vaccination and previous exposure are indeed more capable to block transmission compared to intramuscular vaccination.

7. It is not possible to compare the IM, IN and PI chains, as not all donors were positive in the PI chain. As a result, most sentinel animals also did not get infected, especially in the airborne chain. Therefore, comparing the transmission efficiency as shown in Figure 2D is not correct. This comment is very thoughtful, and we have considered how to best address this. We have decided to also include the data in the figure which shows transmission only if the donor animal was infected. We think that it is also valuable to include the previous version, and have modified the results thus:

Figure 2 C and D:

Figure legend was modified for D. Pie charts summarizing transmission efficiency between naïve, IM vaccinated, IN vaccinated, and PI hamsters across all possible airborne transmission events (left) and events for which the donor animal was confirmed infected (2 out of 5 samples positive by sgRNA qRT PCR (>10copies/rxn) (right)). Number of events is indicated within each pie chart. Pie chart colors: Black = transmission, white = no transmission.

The results read now from Line 180: We compared the airborne transmission efficiency between naïve, IM vaccinated, IN vaccinated, and PI hamsters for transmission events, where the donor animal was confirmed to be positive. We included immunized and naïve sentinels. For naïve hamsters (N = 7 events with an infected donor animal), the airborne transmission efficiency was 71.43% (percentage of all transmission events resulting in an infected sentinel/all transmission events). While IM vaccination reduced of airborne transmission to 40% (N = 10, p = 0.3348, Fisher's exact test, two sided: Odds ratio = 3.75), IN vaccination (N = 6, p = 0.1026,

Fisher's exact test, two sided: Odds ratio = 12.5) reduced it to 16.67 and PI (N = 2, p = 0.1667, Fisher's exact test, two sided: Odds ratio = not calculable) reduced it to 0% (Figure 2 D). It is possible that we did not see infection in some donor animals, because our sampling scheme was not stringent enough. Therefore, we also compared the airborne transmission efficiency using the data across all transmission events. For naïve hamsters, the airborne transmission efficiency was 63%. While IM vaccination reduced of airborne transmission to 29% (p = 1.870, Fisher's exact test, two sided: Odds ratio = 4.167), both IN vaccination (p = 0.0109, Fisher's exact test, two sided: Odds ratio = 21.67) and PI (p = 0.0109, Fisher's exact test, two sided: Odds ratio = 21.67) reduced it to 7%.

To ensure that we present the more conservative assessment (infected donors only), we have also changed the abstract: Line 41: ...(approx. 60%), whereas intranasal vaccination and previous infection displayed a >80% reduction in transmission.

8. Looking at 5dpi/5Dpe Is early to detect a boost of ab response. For the animals that were exposed, it also depends on when transmission occurred.

We agree with the reviewer that 5 dpi/dpe is an early time point. As we were interested in SARS-CoV-2 replication in tissues, it was necessary to select this time point. However, we did observe a boost in humoral response in the challenged donors, and in some sentinel animals. However, we do agree, that this may be more strongly observed at a later time point.

Minor comments

1. Line 66: The authors should be careful with the terminology used here. The intrinsic transmissibility of alpha, delta or omicron was not necessarily higher than that of Wuhan-like viruses, but the spread in the population, which is different. The authors should here add references to original scientific articles, rather than to a general review on Omicron.

We have added references to studies that showed increased transmissibility of D614G (PMID: 33636719), Alpha (PMID: 34545191) or Delta (PMID: 35550680). In the human population Delta transmitted better compared to Alpha (PMID: 35412379, PMID: 35480627). See Line 67.

2. Line 75: what other evolutionary pressures are the authors referring to? What do the authors mean directionality of SARS-CoV-2 evolution? Genetic evolution? Antigenic evolution? The directionality of SARS-CoV-2 antigenic evolution will be only influenced by pre-existing immunity.

SARS-CoV-2 evolution is a function of two pressures: increased transmission and antigenic escape. In the case of SARS-CoV-2 it is not clear how much the first variants were influenced by host immune factors, taking D614G as an example. Data support the hypothesis that early in the pandemic evolution was driven by increased transmissibility. Over time evolutionary change then also became a function of antigenic escape, as herd immunity increased with time (PMID: 33184236, PMID: 32931734).

We have reworded the sentence to: Line 74: To better understand SARS-CoV-2 evolution, it will be crucial to differentiate between two separate evolutionary pressures: increasing transmissibility and antigenic escape.

3. Line 87: indicate here what vaccine strain is in AZD1222

Done, see Line 87.

4. Line 87: change "N=6 hamsters per group", to "Six hamsters per group" at the beginning of

the sentence.

Done, see Line 87.

5. Figure 1: statistical tests described in the text and in the figure legend are not the same. Precise in the legend that A shows data from the lungs and B from the nasal turbinates.

This has been addressed and corrected. The test stated in the text is correct.

6. Line 93: On viral RNA was detected, which does not allow to conclude about viral replication.

We respectfully disagree. We measured sgRNA, which is acknowledged in the field as a surrogate for virus replication, but not infectious virus, for SARS-CoV-2. To ensure this is clear in the text, we have added this statement: Line 93: We measured sgRNA, which is a surrogate for virus replication ^{1,2}, quantity in nasal turbinates and lungs at day 5 post challenge.

1. Singanayagam A, Patel M, Charlett A, et al. Duration of infectiousness and correlation with RT-PCR cycle threshold values in cases of COVID-19, England, January to May 2020. *Eurosurveillance*. 2020;25(32):2001483.

2. Bravo MS, Berengua C, Marín P, et al. Viral Culture Confirmed SARS-CoV-2 Subgenomic RNA Value as a Good Surrogate Marker of Infectivity. *Journal of Clinical Microbiology*. 2022;60(1):e01609-21. doi:doi:10.1128/JCM.01609-21

7. Line 114: change “respectably” for “respectively”

Done see Line 115.

8. Figure 1G: adjusting the white balance would help increasing the contrast and readability of the stainings.

Done.

9. Figure 1E: how was the NP reactivity quantified?

We thank the reviewer for pointing this out. A sentence has been added to the methods. Line 556: Histopathological assessment was performed by a board-certified, blinded pathologist. Nucleoprotein reactivity was assessed by the pathologist.

To the legends we added additional information to specify: Line 720: Nucleoprotein reactivity score: 0 = none, 1 = rare/few, 2 = scattered, 3 = moderate, 4 = numerous, 5 = diffuse.

To the supplement we added: Line 114: **Table S1**: Pathological assessment of IN, IM vaccinated or PI Syrian hamsters on day 5 post challenge. nsf = no significant findings. y = yes. n = no. Nucleoprotein reactivity score: 0 = none, 1 = rare/few, 2 = scattered, 3 = moderate, 4 = numerous, 5 = diffuse.

10. Figure 1D: describe the box plots as in other sub-legends

Added Line 767: D. Lung weights (lung:body weight ratio). Whisker-plots depicting median, min and max values, and individual values, Kruskal-Wallis test, followed by Dunn’s multiple comparison test.

11. Line 717: “measured” instead of “measure”

Done, see Line 774.

12. Figure 1: spell out the abbreviations in the legend.

Abbreviations have been added.

13. Line 129: “To establish the ability and limitations of the naïve Syrian hamster to model transmission”, hamsters do not model transmission, but the hamster transmission model.

Changed to Line 134: “To establish the ability and limitations of transmission over multiple successful rounds through the air and through contact in the Syrian hamster model, we...”

14. Figure 1A is not very informative. It is difficult to understand how the transmission chains were performed. Perhaps a timeline would be easier to understand the different chains of transmission.

This is accurate and we have modified Figure 2 A to look like this and include a timeline:

15. Line 145: add a coma after SARS-CoV-2

Done.

16. Line 152: “If animal was considered infected if 2 out of 5 samples (either a swab, nasal turbinates, or lung tissue sample) had detectable sgRNA, all IM vaccinated donors became infected.” Change to “Animals were considered infected if 2 out of 5 samples (either a swab, nasal turbinates, or lung tissue samples) had detectable sgRNA. All IM vaccinated donors became infected”

Done, see Line 157: Animals were considered infected if 2 out of 5 samples (either a swab, nasal turbinates, or lung tissue sample) had detectable sgRNA (>10 copies/reaction (rxn)). All IM vaccinated donors became infected. In contrast, 5 out of 6 donors in the IN vaccinated group, and 2 out of 6 donors in the PI group became infected (Figure 2 C).

17. Figure 2B: The scheme is unclear. Although it is indicated in the legend that the colors are described on the right of the figure, it is not the case. In the previously infected group, not clear

why the yellow animals (which were the donors in the infection) were then the donors in the subsequent contact/airborne transmission, rather than the kaki ones.

This has been addressed:

We have made edits to the figure legends that make clear which colors refer to what. Line 793: B. Transmission efficiency in hamsters with pre-existing immunity. Hamsters were either vaccinated IM (dark blue) or IN (light blue) against Lineage A or experienced a previous infection with Delta through contact exposure to IN inoculated hamsters (yellow).

18. line 162: authors refer to the chain of transmission but only describe the first transmission round. It is confusing.

We hope that we have understood the reviewer's concerns correctly and have changed the text to read as follows for clarity from Line 173: Due to the importance of airborne transmission, we decided to take two airborne transmission experiments per group out to sentinels 3...

19. Line 166-167. The sentence is confusing. "the increased reduction in airborne transmission already observed between donors and sentinels 1? The virus did not transmit to the donors.

This has been edited for clarity by removal of the confusing wording.

20. Line 756: Animals were not vaccinated with Delta, but with Wuhan.

For clarity we have specified that vaccination was against Lineage A.

21. Figure 3A: which spike was used to detect IgG ab?

While this information is also provided in the methods, we have included that this is against Lineage A spike in the legend. Line 816: A. Anti-spike IgG response (Lineage A spike),...

22. Figure 3B: which quantiles are indicated?

25th and 75th. We have added this information in the legend. Line 818.

23. Figure 3C: not clear why there are so few points in the IM group.

There was no detectable neutralization, and 0 values do not plot onto a log scale. We have now added a new supplemental table which summarizes all raw serology data, see Supplemental Table 3.

24. Figure 3D: are all sentinel animals included here, also those which did not get infected?

All sentinel 1 were included. This figure was changed to include more extensive data for sentinel 1 animals.

25. Figure 4A: not clear what represent the three squares, different days? Although in the legend it is indicated that percentages in the tissues are also shown. In some occasions, there were two animals in chain 2 and in some occasion two animals in chain 3, it is unclear why this was the case.

We hope to have understood the reviewer correctly and assume this confusion stems from the nature of the grey boxes. It is indicated under Figure A and B that the three squares are representations of the days post exposure sampled. We have now changed the location of this label. We have added the information provided for 4B also for 4A: Colors refer to legend on right (D = donor, S = sentinel, NC = naïve control), grey = no sgRNA present in the sample or sequencing unsuccessful. See Line 852 onwards.

26. Figure 4B: what mean the C and O in the purple squares? It is not explained in the legend.

We are unsure what the reviewer is referring to here. There are no “C”s evident to us. “O” indicates a zero value, as in no Delta was detected.

27. Figure 4C and 4D: error bars are sometimes up and sometimes down.

We thank the reviewer for noticing this and have changed them all to match.

Reviewer #3

The authors evaluated the competitive transmission dynamics of Delta and Omicron in Syrian hamsters that were previously (1) immunized intramuscularly (IM) with AZD1222, (2) immunized intranasally (IN) with AZD1222, or (3) recovered from a prior infection (PI) by Delta. Experiments were conducted in parallel with naïve hamster controls. Overall, this is a valuable study and the results are generally sound.

We thank the reviewer for their time and their comments and hope to have addressed them below.

Specific comments:

1. Figure 2D. The denominator for IM, IN and PI groups contained both naïve hamsters and immuned hamsters. I wonder if you can calculate the infection rate separately? If focusing on immuned hamsters, Also you still see a significant difference between the groups (lines 177-181).

Unfortunately, the sample size in animal experiments is often not as high as one would achieve in in vitro work. The reviewer’s question is very important, and we have calculated the statistics if only immune animals are considered. Even though we still compare 5/8 (naive) transmission

events with 2/7 (IM) and 1/7 (IN and PI), we then do not see significance. $P > 0.999$ and $P = 0.1189$, respectively.

2. According to Figure 3, PI hamsters showed higher neutralizing antibody response than IN or IM vaccinated hamsters. Was the difference significant?

Yes. When using a two-way ANOVA, followed by Tukey's multiple comparisons test, we find that:

Tukey's multiple comparisons test	Mean Diff.	95.00% CI of diff.	Below threshold?	Summary	Adjusted P Value
IM vaccinated vs. IN vaccinated	-24.38	-53.87 to 5.115	No	ns	0.1256
IM vaccinated vs. PI	-160.3	-189.8 to -130.8	Yes	****	<0.0001
IN vaccinated vs. PI	-135.9	-165.4 to -106.4	Yes	****	<0.0001

We have added this information to the figure, the legend (Line 820): To assess differences between Delta and Omicron: Two-way ANOVA, followed by Šídák's multiple comparisons test. To assess differences between groups of pre-existing immunity: Two-way ANOVA, followed by Tukey's multiple comparisons test. $N = 16$

Results: Line 233 onwards: Neutralizing antibody titers were highest in the PI group, which neutralized Delta >10-fold better than Omicron ($p < 0.0001$, $N = 16$, two-way ANOVA followed by Šídák's multiple comparisons test) (Figure 3 C). In the IN vaccinated hamsters, 9 out of 16 animals showed no neutralizing antibodies against the Omicron variant. Of IM vaccinated hamsters, 14 out of 16 had no neutralization of the Delta variant and 15 out of 16 had no neutralization of the Omicron variant. Consequently, virus neutralizing capacity was significantly higher in the PI group as compared to the IM and IN vaccinated groups ($p < 0.0001$, $N = 16$, two-way ANOVA followed by Tukey's multiple comparisons test).

3. Would you comment on the dose of AZD1222 that was used to vaccinate hamsters?

The dosage is mentioned in the methods, 2.5×10^8 IU/animal. This dosage has been used in past publications for the Syrian hamster (e.g., Line 398).

4. How many days (provide a range if possible) post-vaccination or post-infection were the hamsters exposed to the donors.

The vaccination and exposure were conducted for all donors and sentinels in the same time frame. The reviewer is correct in pointing out, that therefore there is an additional delay when comparing the inoculation of donors and the exposure of sentinels. It was at least 35 days and maximum 50 days post the vaccination or previous exposure. We have added this to the manuscript methods. Line 415: Transmission: The transmission chains were conducted at least 28 days post vaccination or previous infection (approximate time range: 35 days to 50 days). Line 155: Seroconversion was assessed at least 21 days after. Six animals (donors) were then challenged after at least 35 days (Delta : Omicron mixture at a 1:1 ratio, total of 104 TCID₅₀).

REVIEWERS' COMMENTS

Reviewer #1 (Remarks to the Author):

This revision by Port et al is much improved and is appropriate for publication. In particular, the authors are to be commended for strengthening the clarity throughout the manuscript and for conducting additional experiments where possible; it is understandable that additional transmission experiments and/or experiments that require samples that were not stored cannot be done.

I have a few very minor suggestions on some of the changes.

Fig. 1G – perhaps the length of the scale bar should be defined in the legend.

Fig. S2 – the P-value seems to have been split across two lines for IFN γ and IL-4.

NGS of RBD – perhaps the actual primers that were used should be listed, or a reference provided?

Reviewer #3 (Remarks to the Author):

Comments on "Infection- or AZD1222 vaccine mediated immunity reduces SARS-CoV-2 transmission, but increases competitiveness of Omicron in hamsters"

I thank the authors for addressing my questions and comments. The revised manuscript has greatly improved its clarity. I would appreciate to follow up on a few minor points.

Response to comment #4:

The authors mentioned that the transmission chains were conducted at least 28 days post vaccination or previous infection (approximate time range: 35 days to 50 days) (Line 415). It was not clear what does "approximate time range" mean; should the time range be 28 days (at least 28 days) to 50 days?

Line 287. "In a few hamsters with pre-existing immunity, Omicron was the dominant variant (Table S2). Would the authors please provide a clear count of the number of (donor vs. sentinel/ vaccine type status) animals showing Omicron being the dominant variant?"

Line 290-293. "Omicron sequences in swab samples from the naïve animals (<2%), Omicron was more prevalent in swab samples from hamsters with pre-existing immunity: donors: IM vaccinated = 2.4%, IN vaccinated = 8.7%,". Can the authors confirm if the Omicron detection frequencies in different groups are significantly different?

Line 365-369. "Our findings align with observations from another study... (ref#48)". Reference 48 reported detection of Omicron as the dominant variant from vaccinated index animals (donors) and from naïve hamsters exposed to vaccinated index. In contrast to the findings reported by Reference 48, Figure 4 clearly showed the dominance of Delta over Omicron in donors or sentinels that were previously vaccinated or infected. To avoid confusion, it may be better to revise this sentence.

REVIEWERS' COMMENTS

Reviewer #1: This revision by Port et al is much improved and is appropriate for publication. In particular, the authors are to be commended for strengthening the clarity throughout the manuscript and for conducting additional experiments where possible; it is understandable that additional transmission experiments and/or experiments that require samples that were not stored cannot be done. I have a few very minor suggestions on some of the changes. Fig. 1G – perhaps the length of the scale bar should be defined in the legend. Fig. S2 – the P-value seems to have been split across two lines for IFNg and IL-4. NGS of RBD – perhaps the actual primers that were used should be listed, or a reference provided? Reviewer #3: Comments on “Infection- or AZD1222 vaccine mediated immunity reduces SARS-CoV-2 transmission, but increases competitiveness of Omicron in hamsters” I thank the authors for addressing my questions and comments. The revised manuscript has greatly improved its clarity. I would appreciate to follow up on a few minor points. Response to comment #4: The authors mentioned that the transmission chains were conducted at least 28 days post vaccination or previous infection (approximate time range: 35 days to 50 days) (Line 415). It was not clear what does “approximate time range” mean; should the time range be 28 days (at least 28 days) to 50 days? Line 287. “In a few hamsters with pre-existing immunity, Omicron was the dominant variant (Table S2). Would the authors please provide a clear count of the number of (donor vs. sentinel/ vaccine type status) animals showing Omicron being the dominant variant?” Line 290-293. “Omicron sequences in swab samples from the naïve animals (<2%), Omicron was more prevalent in swab samples from hamsters with pre-existing immunity: donors: IM vaccinated = 2.4%, IN vaccinated = 8.7%,”. Can the authors confirm if the Omicron detection frequencies in different groups are significantly different? Line 365-369. “Our findings align with observations from another study... (ref#48)”. Reference 48 reported detection of Omicron as the dominant variant from vaccinated index animals (donors) and from naïve hamsters exposed to vaccinated index. In contrast to the findings reported by Reference 48, Figure 4 clearly showed the dominance of Delta over Omicron in donors or sentinels that were previously vaccinated or infected. To avoid confusion, it may be better to revise this sentence.	We thank the reviewer for their time and dedication to detail. We hope to have addressed all comments. This has been added. This has been fixed. The primers were added. nCoV_Spike_76L_alt3: GGGCAAAGTGGAAAGATTGCTGA nCoV_Spike_76R_alt0: ACCTGTGCCTGTAAACCATTGA We thank the reviewer for their time and their comments. We hope to have answered all remaining questions. We have edited this and specified exact time ranges: L574: Transmission: The transmission chains were conducted at least 28 days post vaccination or previous infection (infection of donor animals occurred between days 34 and 51, exposure of sentinels occurred between days 35 and 52). We have added this information in L315: In three hamsters with pre-existing immunity, Omicron B.1.1.529 was the dominant variant (Table S2): day 2 swab of one IM vaccinated contact sentinel, days 2 and 3 swabs of one PI donor, and day 2 swab of a second PI donor. This was not significant. We have added this information in L321 We thank the reviewer for this comment and have added: “While we could not report Omicron as the dominant variant in most of our animals with pre-existing immunity, our findings align with observations from another...”
---	---